# PilY1 and minor pilins form a complex priming the type IVa pilus in *Myxococcus xanthus*

Anke Treuner-Lange [1], Yi-Wei Chang [2,3], Timo Glatter[1], Marco Herfurth[1], Steffi Lindow[1], Georges Chreifi[2], Grant J. Jensen [2,4] & Lotte Søgaard-Andersen [1✉]

Type IVa pili are ubiquitous and versatile bacterial cell surface filaments that undergo cycles of extension, adhesion and retraction powered by the cell-envelope spanning type IVa pilus machine (T4aPM). The overall architecture of the T4aPM and the location of 10 conserved core proteins within this architecture have been elucidated. Here, using genetics, cell biology, proteomics and cryo-electron tomography, we demonstrate that the PilY1 protein and four minor pilins, which are widely conserved in T4aP systems, are essential for pilus extension in *Myxococcus xanthus* and form a complex that is an integral part of the T4aPM. Moreover, these proteins are part of the extended pilus. Our data support a model whereby the PilY1/ minor pilin complex functions as a priming complex in T4aPM for pilus extension, a tip complex in the extended pilus for adhesion, and a cork for terminating retraction to maintain a priming complex for the next round of extension.

[1] Max Planck Institute for Terrestrial Microbiology, Karl-von-Frisch Str. 10, 35043 Marburg, Germany. [2] Division of Biology and Biological Engineering, California Institute of Technology, 1200 E California Boulevard, Pasadena, CA 91125, USA. [3] Department of Biochemistry and Biophysics, Perelman School of Medicine, University of Pennsylvania, 422 Curie Boulevard, Philadelphia, PA 19104, USA. [4] Howard Hughes Medical Institute, California Institute of Technology, Pasadena, CA 91125, USA. ✉email: sogaard@mpi-marburg.mpg.de

Type IVa pili (T4aP) are filamentous cell surface structures found in many bacteria and are important for virulence, biofilm formation, adhesion to host cells and abiotic surfaces, motility, DNA uptake, protein secretion, and surface sensing[1,2]. Key to this versatility is their ability to undergo cycles of extension, surface adhesion, and retraction[2]. In Gram-negative bacteria, the T4aP machine (T4aPM) that underlies T4aP formation and function consists of at least ten conserved core proteins

that localize to the outer membrane (OM), the periplasm, the inner membrane (IM), and the cytoplasm (Fig. 1a). All core proteins are important for T4aP extension except for PilT, which is only important for retraction[2]. T4aP can be several microns in length and are mainly composed of the major pilin subunit[2]. The architecture of the T4aPM with and without an extended pilus (namely, piliated and non-piliated forms, respectively) has been determined by cryo-electron tomography (cryo-ET) in *Myxococcus xanthus*

**Fig. 1 PilY1 and minor pilins are essential for T4aP-dependent motility. a** Architectural model of T4aPM. The OM pore (PilQ secretin and TsaP) connect to PilP. PilP connect to a lower periplasmic ring (globular domains of PilN and PilO), which connect to a cytoplasmic ring (PilM)[3]. PilN/O and PilM generate a cage, which is occupied by PilC[3]. PilB and PilT associate with PilC in a mutually exclusive manner during extension and retraction, respectively[3,68,69]. For simplicity, PilB and PilT are not shown separately. Bent arrows, incorporation and removal from the pilus base of PilA during extension and retraction, respectively. Proteins labeled with single letters have the Pil prefix. **b** Schematic of *M. xanthus* gene clusters encoding PilY1 and minor pilins. Numbers below genes, distances between genes. Black lines below, genes deleted in cluster deletion mutants. **c** PilY1 and minor pilins are essential for T4aP-dependent motility. Scale bar, 1mm. Strains motile by T4aP generate flares at the colony edge while strains non-motile by T4aP generate smooth-edged colonies. **d** Minor pilins and PilY1 are essential for T4aP formation. T4aP sheared off from ~15 mg cells were separated by SDS-PAGE and probed with α-PilA antibodies (top rows). Middle row, 40 μg of protein from total cell extracts were separated by SDS-PAGE and probed with α-PilA antibodies (middle rows) and, after stripping, with α-LonD antibodies as a loading control (bottom rows). **e** Minor pilins and PilY1 are essential for T4aP extension. Samples were prepared as in **d**. For better comparison, only 5% of T4aP sheared from the hyper-piliated Δ*pilT* strain (asterisk) and 25% of protein (hash) was applied. **f** Accumulation levels of T4aPM proteins. Protein from $3 \times 10^8$ cells was loaded per lane. In lane labeled Δ, protein from relevant in-frame deletion mutants was loaded. Gaps between lanes indicate lanes that were deleted for presentation purposes. **g** Bipolar localization of mCherry-PilM as a read-out for assembly of T4aPM. Scale bar, 5 μm. The results for the deletions of *fimUpilVpilW en bloc* in **c–e** were previously published[3] and are included for comparison.

and *Thermus thermophilus* and demonstrated that the piliated and non-piliated forms are multilayered structures that span the entire cell envelope[3,4] (Fig. 1a). The PilB ATPase associates with the base of the T4aPM and stimulates pilus extension whereby major pilin subunits are extracted from the IM and inserted at the base of the growing pilus. Conversely, during retraction, the PilT ATPase associates with the base of the T4aPM and stimulates removal of pilin subunits from the pilus base and their reinsertion into the IM[2] (Fig. 1a). Cryo-ET imaging of T4aPMs in *M. xanthus* lacking specific proteins or containing proteins fused to a fluorescent protein allowed to map the location of the ten core proteins within the two forms[3] (Fig. 1a). Importantly, the piliated T4aPM contains a long stem that originates at PilC in the IM, traverses the periplasm, passes through the OM PilQ pore, is continuous with the extracellular pilus, and represents the IM-anchored T4aP. By contrast, the non-piliated T4aPM exhibits a short stem that also originates at PilC in the IM but only extends to the lower periplasmic ring[3] (Fig. 1a). In addition to the ten core T4aPM proteins, the conserved minor pilins and PilY1 protein are also important for T4aP formation. Here we address the function of minor pilins and PilY1 in the *M. xanthus* model system.

Minor pilins, termed minor because of their much lower abundance compared to the major pilin[5], and major pilin are synthesized with an N-terminal type III signal peptide, which is cleaved off by the PilD prepilin peptidase between the Gly and Phe residue of the consensus GFxxxE motif[6]. The mature minor pilins and major pilin share overall sequence characteristics and comprise an N-terminal α-helix followed by a more diverse globular domain[7,8]. The N-terminal segment of the N-terminal α-helix of the major pilin anchors the protein in the IM before it is incorporated into the pilus[2] and also makes up the center of the assembled pilus[2,9]. Minor pilins in *Pseudomonas aeruginosa* and *Neisseria gonorrhoeae* have been detected in purified pili[5,8,10] and in the periplasm[11]; however, it is unclear where they locate within a pilus and whether they are incorporated as a complex or individually. Minor pilins are essential for T4aP extension in *P. aeruginosa*[10] and *M. xanthus*[3] and important for T4aP formation in *N. gonorrhoeae*[8] and *Neisseria meningitidis*[12]. Minor pilins have been suggested to prime T4aP extension, counteract retraction, regulate the extension/retraction balance, and/or promote surface adhesion[3,5,8,12]. Also, formation of the short stem in the non-piliated T4aPM in *M. xanthus* depends on the major pilin as well as on minor pilins[3]. Distinct from minor pilins, PilY1 is a soluble protein with a type I signal peptide followed by a poorly conserved N-terminal region and a well-conserved C-terminal PilY1 domain that forms a modified beta-propeller structure[13]. PilY1 has been reported in the IM[14], the OM[15,16], on the cell surface[17], in purified pili[8,10,18,19], at the pilus

tip[20], and in culture supernatants[14]. In *P. aeruginosa*, *N. gonorrhoeae*, and *N. meningitidis*, PilY1 is important for T4aP formation[10,18,21,22] and PilY1 has been suggested to prime T4aP formation[10] and/or promote host cell adhesion during infections[16,18,20]. Minor pilins and PilY1 are often encoded by clustered genes[1] and have been suggested to form a complex important for T4aP extension in *P. aeruginosa*[10]; however direct experimental evidence for their localization within the T4aPM and for direct interactions between minor pilins and PilY1 is missing.

Here we explore the structure, function, location, and interaction of PilY1 and the minor pilins using the *M. xanthus* T4aP model system and provide evidence that they form a complex, which is an integral part of the T4aPM and functions as a priming complex for pilus extension. In addition, our data support a model in which this complex is present at the tip of the extended pilus for adhesion and that it may function as a cork for terminating retraction.

## Results and discussion

**Lack of PilY1 or minor pilins blocks pilus extension.** We previously identified three gene clusters (cluster_1–_3; numbering also used as suffix for proteins from a cluster) in *M. xanthus* that encode minor pilins[3]. Further analysis here ("Methods") revealed that each cluster encodes a PilY1 protein and four minor pilins (FimU, PilV, PilW, PilX; Fig. 1b). The major pilin PilA and the 12 minor pilins all have the type III signal peptide (Supplementary Fig. 1). Similarly to the PilX homologs of *P. aeruginosa*[19] and *N. gonorrhoeae*[8], the three *M. xanthus* PilX homologs do not contain the conserved Glu5 residue in the processed, mature protein (Supplementary Fig. 1). All 13 mature pilins are predicted to contain the N-terminal α-helix (Supplementary Fig. 1).

Each *M. xanthus* PilY1 protein has a type I signal peptide suggesting their translocation across the IM to the periplasm, and all three contain the conserved PilY1 domain in their C-terminus (Supplementary Fig. 2a, b). Consistently, based on the structure of the PilY1 domain of the *P. aeruginosa* PilY1 protein (3HX6)[13], homology models could be constructed for the PilY1 domain of all three *M. xanthus* PilY1 proteins (Supplementary Fig. 2c). Similarly to *P. aeruginosa* PilY1, PilY1.3 contains an N-terminal von Willebrand factor A (vWFA) domain (Supplementary Figs. 2a and 3a, b) and a homology model could be constructed for this domain based on vWFA domain of the *Catenulispora acidiphila* protein Caci_2163 (4FX5) (Supplementary Fig. 3b). By contrast, PilY1.1 and PilY1.2, similarly to the two *N. gonorrhoeae* PilY1 proteins, do not contain recognizable domains in their N-terminus (Supplementary Fig. 2a). Because *M. xanthus* has a single major pilin, these analyses suggest that the 12 minor pilins

and 3 PilY1 proteins function together with 1 major pilin and a single T4aPM in *M. xanthus*.

Systematic deletions of the various complete gene clusters demonstrated that cluster_1 or cluster_3 alone is sufficient for T4aP-dependent motility (Fig. 1c). Consistently, we observed using a T4aP shear-off assay in which pili are sheared of the surface of cells that cluster_1 or cluster_3 alone is sufficient for T4aP formation (Fig. 1d). Moreover, within each of these two clusters, PilY1 and the minor pilins are essential for T4aP-dependent motility (Fig. 1c) and T4aP formation (Fig. 1d). In the four non-piliated triple deletion mutants (in which genes for all homologs of one or more minor pilins or PilY1 were deleted), further deleting the *pilT* gene for the retraction ATPase PilT did not restore T4aP formation (Fig. 1e). Thus lack of minor pilins and/or PilY1 cause a T4aP extension defect rather than hyper-retraction. The four triple mutants accumulated the ten core T4aPM proteins (Fig. 1f), indicating preserved interactions among the core proteins and preventing protein degradation. In the rod-shaped *M. xanthus* cells, T4aPMs specifically assemble at the two cell poles[3,23–25] and this assembly depends on the PilQ secretin in the OM in *M. xanthus*[23]. Using bipolar localization of an active PilM-mCherry fusion (Fig. 1a and Supplementary Fig. 4a–c) as a read-out for T4aPM assembly at the two poles[23], we observed that PilM-mCherry localized in a bipolar pattern in the four triple mutants but not in the Δ*pilQ* mutant (Fig. 1g) supporting that these mutants still assemble T4aPM. Taken together, we conclude that lack of minor pilins and/or PilY1 blocks pilus extension but not T4aPM assembly.

Furthermore, we performed label-free quantitative mass spectrometry (LFQ-MS) on whole-cell extracts to address which minor pilins and PilY1 proteins accumulate under standard laboratory conditions. We detected all 10 T4aPM core proteins, all cluster_3 proteins, 2–5 cluster_1 proteins, but not cluster_2 proteins (Supplementary Fig. 5). Based on intensity-based absolute quantification (iBAQ), which provide a proxy for absolute protein levels ("Methods"), these analyses also revealed that PilA is one of the most abundant proteins in *M. xanthus*, while cluster_3 and cluster_1 proteins were much less abundant as previously reported for minor pilins in *P. aeruginosa*[5]. Because cluster_3 proteins predominated in wild-type (WT) cells and cluster_3 proteins alone were sufficient for T4aP formation and function, we henceforth focused on cluster_3.

In a Δ1Δ2cluster strain, additional deletion of individual cluster_3 genes together with complementation experiments showed that *pilV3*, *pilW3*, *pilX3*, and *pilY1.3* are essential for T4aP-dependent motility and T4aP extension, while T4aP extension was not completely abolished in the Δ*fimU3* mutant in the absence of PilT (Supplementary Fig. 6a–d).

**The major pilin and minor pilins interact**. To address whether the mature forms of the major pilin PilA and minor pilins of cluster_3, i.e., pilins without the type III signal peptide, interact, we used the Bacterial Two-Hybrid (BACTH) system as previously described to detect interactions between minor and major pilins of T4aP systems and the pseudopilins of type II secretion systems (T2SS)[10,26–31]. To this end, the T18 and T25 fragments of the adenylate cyclase were fused to the N-terminus of the full-length mature pilins. We observed strong self-interactions of PilA and PilX3 along with strong interactions between PilA and FimU3, PilV3, PilW3, and PilX3 (Fig. 2a). FimU3, PilW3, and PilX3 interacted in all pairwise combinations while PilV3 only showed a weak interaction with PilA (Fig. 2a). In control experiments for specificity, we observed that neither PilA nor minor pilins fusions interacted with the bitopic IM protein AglQ, a MotA homolog, fused to the C-terminus of the T18 or T25 fragments (Fig. 2a),

while both AglQ constructs interacted with its IM partner protein AglR, a MotB homolog, as previously shown[32] (Supplementary Fig. 7a). Moreover, in fractionation experiments, we observed that all six T18 fusions, and therefore likely also the T25 fusions, were enriched in the membrane fraction (Fig. 2b). T18-PilV accumulated at a much lower level than the remaining fusions, possibly explaining why no interactions were observed to the other minor pilins and only a weak interaction to PilA (Fig. 2a). Finally, a T18 fusion to a PilA variant lacking the N-terminal segment of the N-terminal α-helix that anchors the protein in the IM did not interact with the minor pilins and was not enriched in the membrane fraction (Supplementary Fig. 7b, c). Altogether, these results demonstrate that mature PilA and minor pilins interact and are in agreement with the formation of a complex containing all five pilins.

**PilY1 and minor pilins interact**. Because PilY1.3 constructs did not accumulate in the BACTH, we performed pull-down mass spectrometry experiments to test for interactions between PilY1.3 and cluster_3 minor pilins using a Δ1Δ2cluster *M. xanthus* mutant additionally deleted for *pilY1.3* and synthesizing either a PilY1.3 protein with a C-terminal FLAG tag (PilY1.3-FLAG) or a green fluorescent protein (GFP)-FLAG protein (negative control). Both FLAG-tagged proteins accumulated and PilY1.3-FLAG supported motility, suggesting that the FLAG-tag did not disrupt PilY1.3's functional interactions with other T4aPM components (Fig. 3a, b). Among proteins known to be related to T4aP formation and function, three minor pilins (PilX3, PilW3, and FimU3) and the core T4aPM protein PilO, which is part of the lower periplasmic ring (Fig. 1a), were highly enriched in the PilY1.3-FLAG pull-down experiments (Fig. 3c and Supplementary Fig. 8a). PilA is highly abundant and binds non-specifically to the α-FLAG matrix, precluding its analysis by this method. Altogether, the BACTH experiments and PilY1.3-FLAG pull-down experiments support that PilY1.3 interacts with cluster_3 minor pilins and minor pilins interact with PilA.

If PilY1.3 and minor pilins are integral components of the T4aPM, the prediction is that they would localize in a polar pattern because T4aPM's only assemble at the cell poles in *M. xanthus*. To test this hypothesis, we expressed a *pilY1.3-sfGFP* or *pilW3-sfGFP* fusion gene in a Δ1Δ2 cluster mutant additionally deleted for either *pilY1.3* (strain SA7791) or *pilW3* (strain SA7886). PilY1.3-sfGFP and PilW3-sfGFP, which was used as a proxy for the cluster_3 minor pilins, accumulated but did not support motility (Fig. 3a, d, e). In all, 97% and 86% of cells showed polar localization of PilY1.3-sfGFP and PilW3-sfGFP, respectively (Fig. 3f). The observation that their localization was dependent on PilQ (Fig. 3f), which is essential for T4aPM assembly in *M. xanthus*[23], suggest T4aPM incorporation of PilY1.3-sfGFP and PilW3-sfGFP, even though they cannot support T4aPM function.

We reasoned that if minor pilins and PilY1.3 interact directly, then their stability or incorporation into the T4aPM might be mutually dependent. Indeed, in the absence of all four cluster_3 minor pilins, PilY1.3-sfGFP did not accumulate (Fig. 3d). However, PilY1.3-sfGFP accumulated in strains lacking FimU3, PilV3, or PilW3 but not in a strain lacking PilX3 (Fig. 3d). In the three former strains, polar localization of PilY1.3-sfGFP was significantly reduced (Fig. 3f). Similarly, in the absence of PilY1.3, PilW3-sfGFP accumulation was significantly reduced and the protein did not display polar localization (Fig. 3e, f). Of note, PilY1.3-sfGFP and PilW3-sfGFP accumulated in the absence of PilA and both proteins localized polarly in the absence of PilA (Fig. 3d–f) supporting that they are incorporated into the T4aPM independently of PilA.

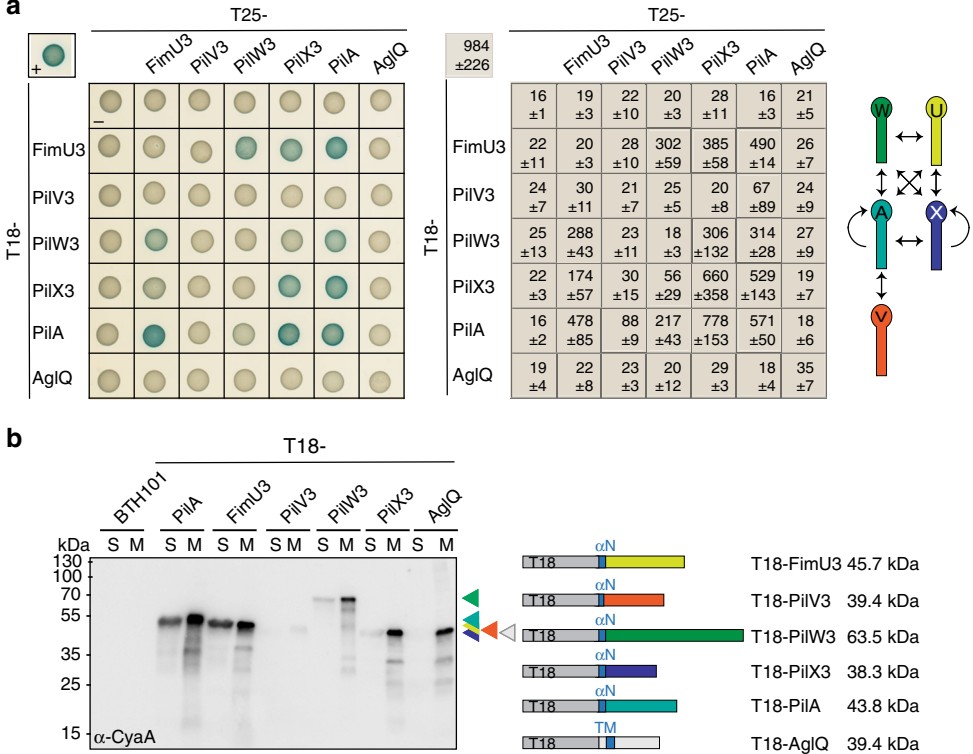

**Fig. 2 PilA and cluster_3 minor pilins interact. a** BACTH analysis of PilA and cluster_3 minor pilin interactions. T25 and T18 were fused to the N-terminus of the indicated mature, full-length proteins. Positive control (plus) in upper left corner, T25-Zip + T18-Zip; T18 and T25, negative control (minus). Full-length AglQ fused to T25 and T18 served as a specificity control. Left panel, representative images of *E. coli* BTH101 expressing the indicated protein fusions. For every tested interaction pair, four clones were tested with similar results. Middle panel, relative activity of β-galactosidase shown as mean ± standard deviation ($n = 4$) for the pairs on the left. Right panel, schematic summarizing observed interactions. **b** Fractionation of BTH101 *E. coli* cells expressing the indicated T18 fusions. Cleared cell lysates were fractionated into soluble (S) and membrane (M) fractions. αN N-terminal α-helix in mature pilins, TM trans-membrane helix in AglQ. Proteins from the same number of cells were loaded per lane. Right panel, schematics of domain organization and calculated molecular sizes of the corresponding proteins.

We performed pull-down experiments to identify PilY1.3-sfGFP and PilW3-sfGFP interaction partners within the T4aPM using the strains SA7791 and SA7886, respectively. Among enriched proteins known to be related to T4aP formation and function (Supplementary Fig. 8b, c, e), PilX3, PilW3, FimU3, and the core protein TsaP were highly enriched in the PilY1.3-sfGFP pull-downs (Fig. 3g), while PilY1.3, PilX3, PilO, and PilB were highly enriched in the PilW3-sfGFP pull-downs (Fig. 3h) supporting direct interactions between minor pilins and PilY1.3 as well as between minor pilins within the T4aPM. Pairwise comparisons of enriched proteins in the PilY1.3-FLAG, PilY1.3-sfGFP, and PilW3-sfGFP pull-down experiments revealed only two additional proteins that were enriched in all three experiments (Supplementary Fig. 8e), both of which are predicted to be cytoplasmic proteins and none of which have been associated with T4aP formation and function. The highly abundant major pilin PilA bound non-specifically to the α-GFP matrix again precluding analysis of PilA in these experiments.

Together, these observations support that (1) PilY1.3 and PilW3 and, therefore, more generally cluster_3 minor pilins are incorporated into the T4aPM, (2) minor pilins and PilY1.3 interact within the T4aPM, and (3) their incorporation into the T4aPM is mutually dependent but independent of PilA. Moreover, the bulky PilY1.3-sfGFP and PilW3-sfGFP proteins are incorporated into the T4aPM but somehow jam the machine.

**PilY1 and minor pilins form a priming complex in the T4aPM.** We used cryo-ET to structurally clarify where minor pilins and

PilY1 localize within the T4aPM. First, we imaged a Δ1Δ2Δ3*pilY1* mutant that lacks all three PilY1 proteins. Consistent with its T4aP extension defect, only non-piliated T4aPM structures were identified in the tomograms. In the subtomogram average, this mutant structure lacked most of the short stem and a "plug" density in the mid-periplasmic ring at the entry to the PilQ secretin vestibule (Fig. 4a, second column). Also, the lower periplasmic ring, which is composed of the globular domains of PilN and PilO[3] (Fig. 1a), was perturbed as previously observed in the Δ*pilA* mutant and in a Δ1Δ2Δ3*fimUpilVpilW* mutant[3] (henceforth, Δ9 minor pilin mutant) (Fig. 4a, third and fourth columns). Moreover, similar to the Δ1Δ2Δ3*pilY1* mutant, the subtomogram average of the Δ9 minor pilin mutant lacked most of the short stem and the plug density (Fig. 4a, fourth column). By contrast, the subtomogram average of the Δ*pilA* mutant mostly lacked the lower part of the short stem while the upper part and the plug remained intact (Fig. 4a, third column). These observations are in agreement with PilA, minor pilins, and PilY1 accounting for the short stem and plug densities. Because the Δ*pilA* mutant incorporates the remaining core proteins[3,23], minor pilins (using PilW3-sfGFP as a proxy) as well as PilY1.3-sfGFP into the T4aPM (Fig. 3f), these data also support that the lower part of the short stem is accounted for by PilA while the upper part of the short stem as well as the plug are accounted for by the minor pilins and PilY1.3.

Because minor pilins and PilY1.3 are mutually dependent for incorporation into the T4aPM, their arrangement within the T4aPM was less clear. To map these proteins within the T4aPM, we took into account that PilY1.3 interacts with the minor pilins

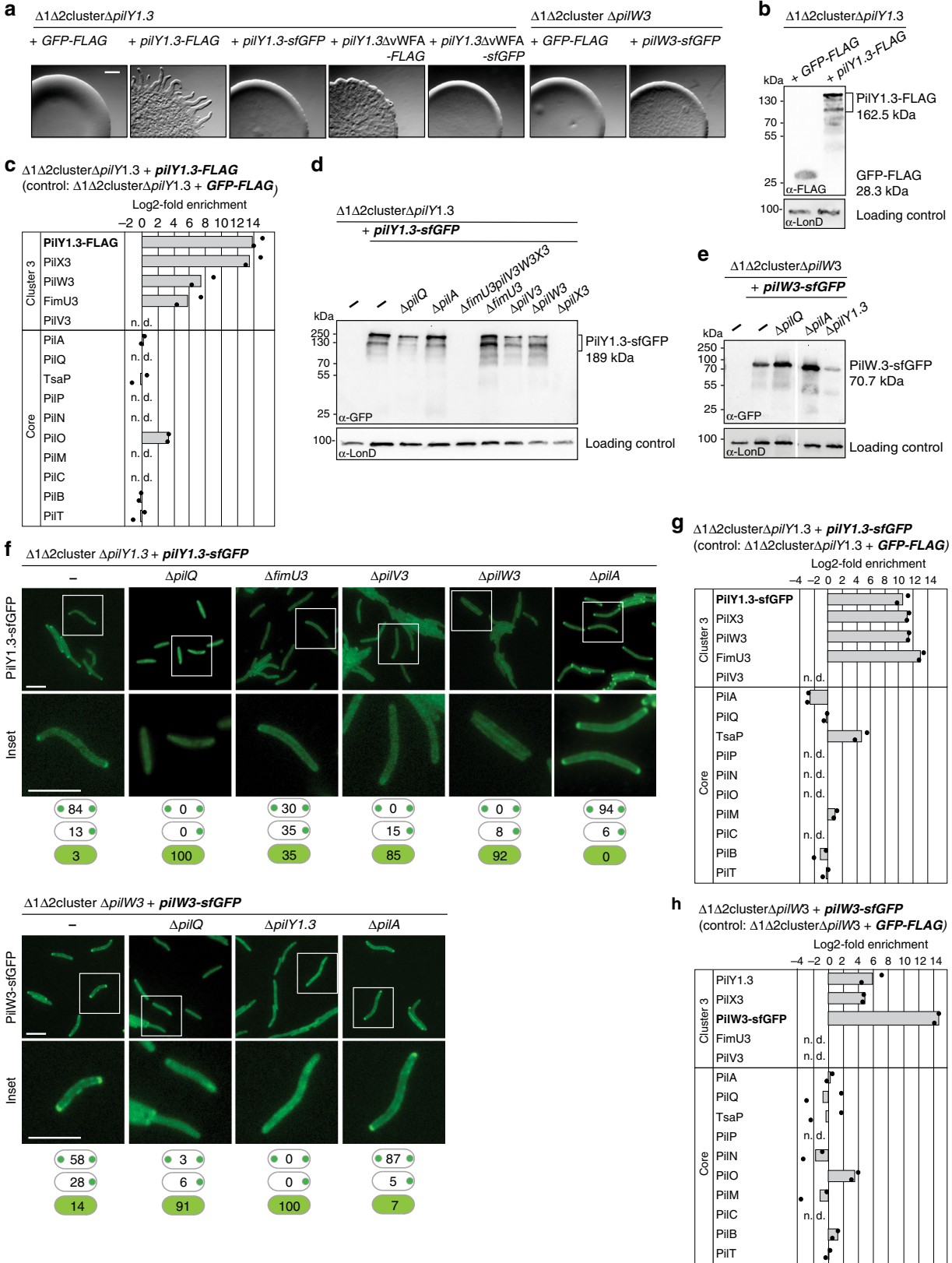

(Fig. 3c, g, h) and the minor pilins interact with PilA (Fig. 2a, b). We first hypothesized that the minor pilins would make up the part of the short stem above PilA, and PilY1 would make up the plug. However, the size of the plug density (~4 nm in diameter; Fig. 4b, first column) is not sufficiently large to fit the entire PilY1.3 protein (molecular weight of ~160 kDa giving a

theoretical size of ~7 nm in diameter if calculated as a single globular protein), thus making this arrangement unlikely. Next, we considered that PilY1.3 also interacts with PilO in the lower periplasmic ring in the pull-down experiments. Because sequence analysis suggests that PilY1.3 contains two globular domains, i.e., the N-terminal vWFA domain and the C-terminal PilY1 domain,

**Fig. 3 Cluster_3 minor pilins and PilY1.3 interact. a** T4aP-dependent motility assays for strains with tagged PilY1.3 or PilW3. Strains of the indicated genotypes were incubated 24 h. Scale bar, 1 mm. **b** Accumulation levels of PilY1.3-FLAG and GFP-FLAG. Protein from $3 \times 10^8$ cells were loaded per lane and blots probed with α-FLAG antibodies and α-LonD antibodies (loading control). **c** PilY1.3 and cluster_3 minor pilins interact. Pull-down experiments with α-FLAG matrix on cell extracts from strain of the indicated genotype expressing PilY1.3-FLAG or GFP-FLAG (negative control). Samples from two biological replicates and two negative controls were analyzed by LFQ-MS; mean iBAQ values and log2-fold enrichment in PilY1.3-FLAG samples compared to GFP-FLAG samples were calculated; enrichment for PilY1.3-FLAG was imputed. Columns represent mean of log2-fold enrichment ($n = 2$); dots represent the corresponding data points; n.d., not detected in PilY1.3-FLAG samples. **d, e** Accumulation levels of PilY1.3-sfGFP and PilW3-sfGFP. Protein from $3 \times 10^8$ cells was loaded per lane and blots were probed with α-GFP antibodies and α-LonD antibodies (loading control). Gap between lanes indicate lanes that were deleted for presentation purposes. **f** PilY1.3-sfGFP and PilW3-sfGFP localize bipolarly in the presence of assembled T4aPM. Insets are enlargements of the boxed areas. Note that the PilW3-sfGFP clusters formed in the ΔpilQ strain were smaller and of lower intensity than in WT. $N > 100$ cells per strain; localization patterns in percentage indicated in schematics (bipolar, unipolar, diffuse (top to bottom)). Scale bars, 5 μm. **g** PilY1.3-sfGFP and cluster_3 minor pilins interact. Pull-down experiments with α-GFP matrix and cell extracts of cells of the indicated genotype containing PilY1.3-sfGFP or GFP-FLAG (negative control). Samples were analyzed as in **c**; enrichment was imputed for PilY1.3-sfGFP. **h** PilW3-sfGFP and cluster_3 proteins interact. Pull-down experiments with α-GFP matrix on cell extracts of the indicated genotype containing PilW3-sfGFP or GFP-FLAG (negative control). Samples were analyzed as in **c**; enrichment was imputed for PilW3-sfGFP.

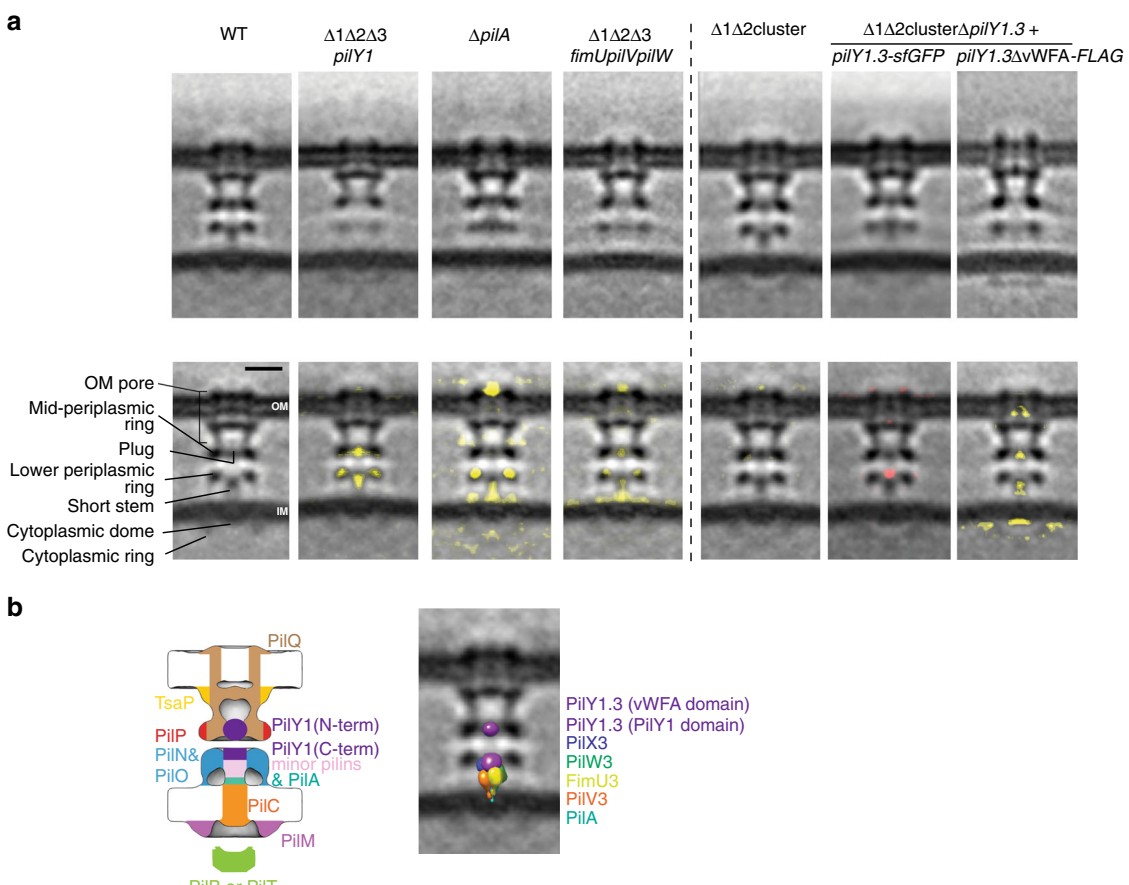

**Fig. 4 PilA, minor pilins, and PilY1 make up the short stem and plug. a** Structures (subtomogram averages) of T4aPM in strains of the indicated genotypes. Top row, central slices through subtomogram averages of empty T4aPM. Lower row, first panel describes densities observed in subtomogram average of empty WT T4aPM (see also Fig. 1a) as described in ref. [3], panels 2–5 show differences between T4aPM in mutants versus the WT (lacking densities, yellow), and panels 6 and 7 the difference between T4aPM in SA7791 (Δ1Δ2cluster ΔpilY1.3 + PilY1.3-sfGFP) and SA8764 (Δ1Δ2cluster ΔpilY1.3ΔvWFA-FLAG) versus T4aPM in Δ1Δ2cluster structure (lacking densities, yellow; extra density, red). Panels 1, 3, and 4 in both rows are from ref. [3] (reprinted with permission from AAAS). Scale bar, 10 nm. **b** Model of T4aPM priming complex. Panel 1, component map of non-piliated T4aPM with priming complex and proteins assigned to the densities in the subtomogram average of the non-piliated WT T4aPM; for simplicity, PilB and PilT are not shown separately. Panel 2, hypothetical structural model of priming complex after low-pass filtering to 3-nm resolution and placed into the subtomogram average of non-piliated WT T4aPM.

separated by regions of undefined structure (Supplementary Figs. 2a, c and 3b), we therefore hypothesized that one PilY1.3 domain could make up the uppermost part of the short stem close to the lower periplasmic ring and the second domain could account for the plug density. To test this hypothesis, we first used

a strategy that previously allowed us to map other core proteins (PilP and PilO) in the T4aPM subtomogram averages[3]. Specifically, we imaged by cryo-ET two Δ1Δ2cluster strains one of which exclusively expressed untagged PilY1.3 and the other exclusively PilY1.3 tagged with a super-folder GFP protein

(PilY1.3-sfGFP; strain SA7791). Importantly, the PilY1.3-sfGFP can be incorporated into the T4aPM but somehow jams the machine for T4aP extension (Fig. 3a, f). We then looked for an additional sfGFP density in the subtomogram average of the T4aPM from the strain containing PilY1.3-sfGFP compared to the strain containing untagged PilY1.3. The subtomogram average of the latter strain (Fig. 4a, fifth column) appeared essentially identical to that of the subtomogram average of WT T4aPM (Fig. 4a, first column), consistent with the observation that cluster_3 proteins predominate in WT cells (Supplementary Fig. 5). As expected, we only observed non-piliated T4aPM in cells of SA7791 (Fig. 4a, sixth column). More importantly, the T4aPM subtomogram average of this strain not only contained the short stem and plug but also an additional density at the tip of the short stem compared to that of the isogenic strain with untagged PilY1.3 (Fig. 4a, compare sixth and fifth column), and this density likely represents the bulky sfGFP tag. Because the sfGFP density is at the tip of the short stem, close to the lower periplasmic ring consisting of PilN and PilO, and the sfGFP tag was fused to the C-terminus of PilY1.3, these data support that PilY1.3's conserved C-terminal PilY1 domain is located at the tip of the short stem in the non-piliated T4aPM structure and that the less conserved N-terminal domain could account for the plug density at the entry to the PilQ periplasmic vestibule. Subsequently, to more precisely determine the orientation of PilY1.3 within the T4aPM, we analyzed two PilY1.3 variants deleted for the N-terminal vWFA domain (PilY1.3ΔvWFA), i.e., variants with either a C-terminal FLAG-tag (PilY1.3ΔvWFA-FLAG) or a C-terminal sfGFP-tag (PilY1.3ΔvWFA-sfGFP) (Supplementary Fig. 9a, b). The two proteins accumulated in strains deleted for cluster_1 and _2 and containing one of these proteins as the only PilY1 protein (Supplementary Fig. 9a, b) but none of the two proteins supported T4aP-dependent motility (Fig. 3a). Importantly, PilY1.3ΔvWFA-sfGFP localized polarly in 84% of cells and this localization was dependent on PilQ (Supplementary Fig. 9c). The polar clusters were smaller and of lower intensity than for PilY1.3-sfGFP (Fig. 3f), suggesting that PilY1.3ΔvWFA-sfGFP is incorporated (although likely less efficiently than full-length PilY1.3-sfGFP) into the T4aPM but cannot support T4aPM function. To identify proteins that interact with PilY1.3ΔvWFA, we performed PilY1.3ΔvWFA-FLAG pull-down experiments. Three minor pilins (PilX3, PilW3, and FimU3) and the core protein PilO in the lower periplasmic ring were highly enriched in these experiments (Supplementary Figs. 9d and 8d). Precisely these four T4aPM proteins are also enriched in the pull-down experiments with full-length PilY1.3-FLAG (Fig. 3c and Supplementary Fig. 8a, d, e). These data not only support that it is the conserved C-terminal PilY1 domain that mediates the interaction from PilY1.3 to the minor pilins and PilO but they also support that it is this domain of PilY1.3 that is located in the short stem close to the lower periplasmic ring. As expected, in tomograms of the PilY1.3ΔvWFA-FLAG mutant only non-piliated T4aPM structures were identified. In the subtomogram average, the T4aPM lacked the short stem and plug densities while the lower periplasmic ring was intact (Fig. 4a, seventh column). Because the lower periplasmic ring depends on PilA, minor pilins, and PilY1 (Fig. 4a, second, third, and fourth columns), the subtomogram average supports that PilA, minor pilins, and PilY1.3ΔvWFA-FLAG are incorporated into the T4aPM. In agreement with the smaller size and reduced intensity of the polar PilY1.3ΔvWFA-sfGFP clusters (Supplementary Fig. 9c), we speculate that the absence of clear densities for the short stem and plug is an indication that the N-terminal vWFA domain in PilY1.3 domain may have an important role in stably incorporating and positioning PilA, minor pilins, and PilY1.3 in the T4aPM. The combination of these observations support a model whereby the

C-terminal PilY1 domain is part of the short stem close to the lower periplasmic ring, and it is in agreement with the notion that the vWFA domain of PilY1.3 accounts for the plug density (Fig. 4b, left panel).

Consistent with the overall assignment of minor pilins and PilY1.3 to the short stem and plug densities, we found that (1) PilY1.3-sfGFP and PilW3-sfGFP accumulated and showed bipolar localization (indicating T4aPM incorporation) in the ΔpilB mutant whose T4aPM structure contains the short stem and the plug densities[3] (Supplementary Fig. 10a–c) and (2) accumulation and/or the size and intensity of polar of PilY1.3-sfGFP and PilW3-sfGFP clusters were reduced in the ΔpilC and ΔpilP mutants whose T4aPM structures lack the short stem and plug densities[3] (Supplementary Fig. 10a–c).

Collectively, these observations are in agreement with a model whereby minor pilins and PilY1.3 are parts of the non-piliated T4aPM and account for part of the short stem as well as the plug at the entry to the PilQ secretin vestibule in *M. xanthus* (Fig. 4b, left panel). They also support that (1) PilQ is required for incorporation of minor pilins and PilY1.3 into the T4aPM but is not sufficient to stably anchor these proteins, (2) PilA is not essential for minor pilin and PilY1 incorporation into the T4aPM, and (3) minor pilins and PilY1.3 are mutually dependent for stability and incorporation into the T4aPM. Because the short stem and plug is part of the T4aPM in the absence of PilB, we conclude that these two structural elements are integral parts of the non-piliated T4aPM and not incorporated into the machine as part of the PilB-stimulated pilus extension process. Also, based on the observation that PilW3 and PilY1.3 may directly interact with PilO (Fig. 3c, h and Supplementary Figs. 9d and 8a, c, d), we speculate that the reduced density in the lower periplasmic ring in the subtomogram averages in the absence of PilA, minor pilins, or PilY1 might be a consequence of the short stem and plug helping to stabilize this ring structure. We also speculate that, in the T4aPM subtomogram average of the ΔpilA mutant (Fig. 4a, third column), the clear gap between the remaining short stem and the IM support that in the absence of PilA the minor pilins likely do not properly interact with one another but instead "float" in the region with their N-terminal α-helix in the IM and globular head domains near the lower periplasmic ring to interact with other T4aPM components including PilY1. Their random locations therefore caused a decrease of the averaged density at the lower part of the short stem. Because PilY1 and minor pilins are essential for T4aP extension, these data also support a model whereby the short stem and plug densities represent a priming complex for T4aP formation composed of, from the IM outwards, PilA, the minor pilins, and PilY1 (Fig. 4b, left).

As a further test of our model for the arrangement of PilA, minor pilins, and PilY1 in the priming complex, we took advantage of available structural information for homologous proteins to generate a hypothetical structural model of the priming complex using the cluster_3 proteins. Of note, the goal of this model was not generating a precise structural model of the priming complex but to test the feasibility of the suggested arrangement of components. Specifically, we used a fragment of five major pilin subunits from the atomic structure of the *N. gonorrhoeae* T4aP filament[9] and replaced the globular domain of the four upper subunits with homology models of the globular domains of *M. xanthus* FimU3, PilV3, PilW3, and PilX3 and the lower subunit with a homology model of the globular domain of *M. xanthus* PilA (Supplementary Fig. 11a). PilV3 only interacted with PilA in the BACTH and was not detected in the PilY1.3 and PilW3 pull-down experiments, yet it is essential for pilus formation. Therefore, our working hypothesis is that PilV3 lies between PilA and the rest of the priming complex and we speculate that its binding to PilA may be strong but relatively

weaker to the rest of the priming complex. Because PilX3 was most highly enriched in the pull-down experiments with functional PilY1.3-FLAG and explicitly important for stabilization of PilY1.3-sfGFP, we placed PilX3 at the top of the five-subunit structure. Because PilW3-sfGFP pulled-down PilX3, we tentatively placed PilW3 below PilX3 followed by FimU3. Next, we separately modeled the structures of the vWFA domain in the N-terminus of PilY1.3 and the conserved C-terminal PilY1 domain (Supplementary Fig. 11a) and associated the latter with the top of the PilA/minor pilin complex, completing the hypothetical structural model of the short stem (Supplementary Fig. 11b). Subsequently, we calculated molecular surfaces on the models and filtered the resolution to 3 nm (corresponding to the resolution of T4aPM subtomogram averages; Supplementary Fig. 11b). The dimensions of the generated structural envelopes of the complex composed of one PilA/four minor pilins/one PilY1 domain and the N-terminal vWFA domain of PilY1.3 fit into and do not exceed the short stem and plug, respectively, in the subtomogram average of WT T4aPM (Fig. 4b, right, and Supplementary Fig. 11c). It is important to note that in this model only 46% of the PilY1.3 sequence is represented due to the lack of structural information about the remaining residues (Supplementary Fig. 2a). We suggest that the unmodeled residues in PilY1.3 are unstructured linkers between the plug and the tip of the short stem, and because of flexibility there is no clear density for them in the subtomogram averages. We conclude that the hypothetical structural model supports the feasibility of the suggested arrangement of components in the T4aPM. The exact

structure of the globular domains of the pilins, their precise order and packing, as well as the stoichiometry of the involved proteins remain to be revealed by future studies.

**PilY1 and minor pilins are present in T4aP.** Because the function of a priming complex would be to serve as a scaffold for initiating PilA incorporation into the growing pilus, this predicts that the priming complex would most likely exit from the T4aPM and localize at the tip of the extended pilus. We also considered that PilY1.3-FLAG supported T4aP-dependent motility (and, therefore, T4aP extension) while PilY1.3-sfGFP and PilW3-sfGFP did not (Fig. 3a) demonstrating that a bulky addition obstructs PilY1.3 and PilW3 function possibly by blocking their translocation through the PilQ secretin pore. To examine whether PilY1.3 and/or minor pilins are incorporated into the pilus, we purified sheared T4aP (Supplementary Fig. 12a) and quantified the associated proteins (Fig. 5a, b). As expected, PilA was highly enriched in purified T4aP; more importantly, three cluster_3 minor pilins and PilY1.3 were also enriched while the remaining T4aPM proteins were not (Fig. 5a), documenting that this enrichment was not due to co-purification of the entire T4aPM. Based on iBAQ values, the absolute abundance of cluster_3 minor pilins and PilY1.3 in purified T4aP was similar, suggesting that they incorporate in the T4aP as a complex rather than as scattered individual proteins (Fig. 5b). The data also reveal a ratio of PilY1–minor pilins to PilA molecules of 0.5–2.0:10,000 in purified pili (Fig. 5b). Because a 10-μm long pilus contains ~10,000 PilA subunits and T4aP in *M. xanthus* are

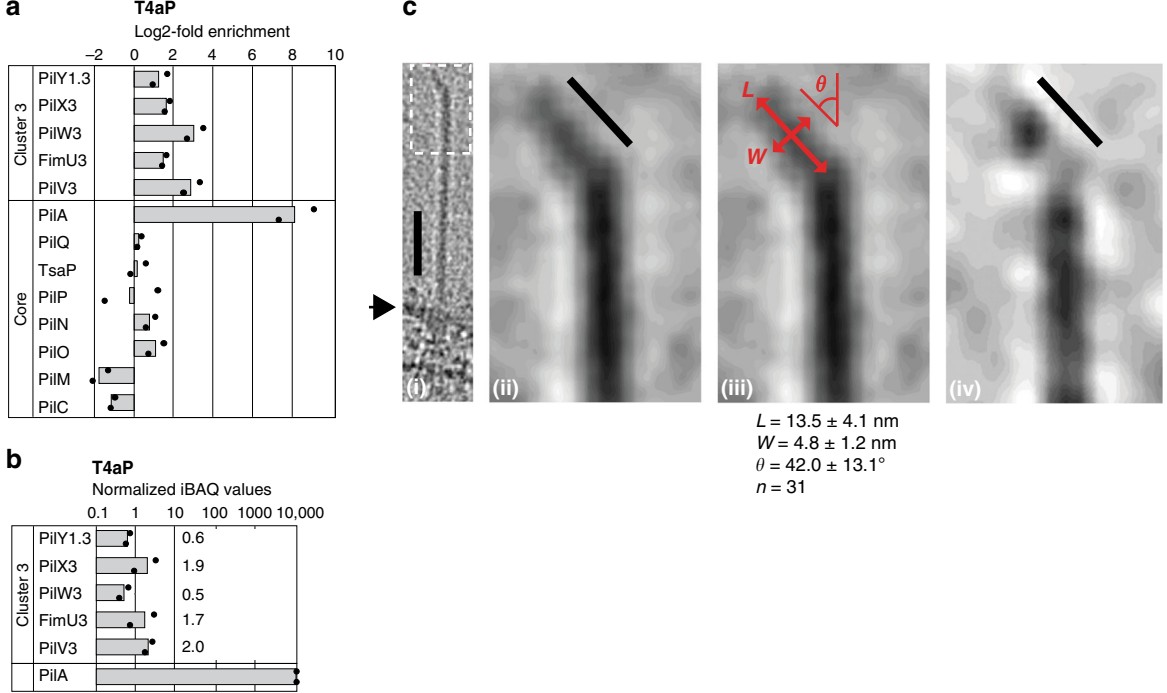

**Fig. 5 Minor pilins and PilY1 are present in T4aP. a** Cluster_3 minor pilins and PilY1.3 are enriched in purified T4aP. T4aP were purified from the hyper-piliated Δ*pilT* mutant and the non-piliated Δ*pilB*Δ*pilT* mutant (negative control). Samples from two biological replicates were analyzed by LFQ-MS, mean iBAQ values were determined, and log2-fold enrichment in Δ*pilT* samples compared to the negative control was calculated. For PilW3, enrichment was calculated using imputed values. Columns represent mean of log2-fold enrichment (*n* = 2), dots represent the corresponding data points. **b** iBAQ values from negative control were subtracted from iBAQ values of the Δ*pilT* samples from **a**; subsequently, these values were normalized relative to an iBAQ value of 10,000 for PilA in the Δ*pilT* sample. The columns represent the mean of normalized IBAQ values (*n* = 2), dots represent the corresponding data points. **c** Cryo-ET images of T4aP tip structures. First panel, T4aP extending from cell surface (arrow) and with the kinked structure at the tip; scale bar, 50 nm. Second panel, magnification of area in stippled box in first panel illustrating kinked structure without a density gap; scale bar, 10 nm. Third panel, the same pilus tip image as in the second panel, with double-headed arrows indicating the directions of length (*L*), width (*W*), and kink angle (*θ*) measurements. The result of measurements on the 31 available T4aP tip is listed underneath the panel (mean ± standard deviation). Fourth panel, kinked tip structure with a density gap between terminal globular density and the tip of the pilus shaft; scale bar, 10 nm.

typically 2–6 µm in length, this molar ratio supports that each T4aP pilus contains only a single PilY1.3/minor pilin complex rather than multiple complexes along the pilus. We note that PilV3 was detected in purified pili (but not in pull-down experiments with PilY1.3 and PilW3) supporting that PilV3 is part of a minor pilin/PilY1.3 complex (Fig. 4b).

To search for a potential tip-localized PilY1.3/minor pilin complex, we performed immunogold labeling and transmission electron microscopy on sheared T4aP and cells from strains expressing the functional PilY1.3-FLAG variant. Pili-associated gold particles were observed for sheared T4aP and, in the case of intact cells, at a distance from the cell surface. Not more than one gold particle per pilus was detected (Supplementary Fig. 12b, c), corroborating that PilY1.3 is present in the extended pili. Because we were unable to unambiguously identify T4aP tips with these methods after extensive trials, we searched instead for T4aP structures in the cryo-tomograms that could be tracked all the way from the T4aPM in the cell envelope to the pilus tip. We identified 43 pili that fulfilled this criterion; these pili were relatively short (50–500 nm) because longer pili could not be tracked from the cell surface all the way to the tip due to the limited field of view of cryo-ET. In all, 72% (31 out of 43) of these pili had a highly specific, low density, kinked structure at the tip (Fig. 5c and Supplementary Fig. 13). After detailed inspection and measurement of the 31 available kinked tip structures, the notable variations of their dimensions and kinked angle (Fig. 5c) prompted us to speculate that the kinked structure could comprise the parts of PilY1.3 with density discontinued to the short stem (i.e., in the plug) and/or invisible in the subtomogram average due to their flexibility (e.g., in between the plug and short stem or above the plug) in the T4aPM. We also speculated that these parts of PilY1.3 would be able to dangle around at the pilus tip after the stem/pilus extends out from the cell resulting in the kinked structures, while in the T4aPM these parts of PilY1.3 would be more aligned because of interactions with the remaining T4aPM components. Indeed, the average size of the kinked tip structure is well capable to accommodate the PilY1.3 vWFA domain, which we have proposed as the plug density (~4 nm in diameter) in the T4aPM (Fig. 5c(iii)). More interestingly, in some of the kinked tip structures a distinct globular density can be seen, and it connects to the tip of the pilus shaft through a lighter density (Fig. 5c(iv)). The size of the globular density and its distance from the tip of pilus shaft match well to the size of the plug in the T4aPM and its distance from the tip of the short stem (Supplementary Fig. 11c, d). Thus these observations are in agreement with the notion that the kinked structure at the pilus tip may represent the priming complex with the PilY1.3 C-terminal domain at the pilus proximal end of the kink and the N-terminal vWFA domain at the ultimate tip (Supplementary Fig. 11c, d). One reason that 30% of the pili tips were seen without a clear kink in the tomograms might be because the flexible tips were by chance aligned with the pilus axis when frozen. At this point, we could not rule out the possibility that some of the pilus tips observed were from snapped pili due to their surface binding and retraction. However, since the cells used for these analyses were cultured in suspension ("Methods") with limited chance for surface binding before being frozen directly for cryo-ET imaging, we speculate that, if there are broken pili, they would only account for a minority of pili tips in the whole population and not represent the kinked structure in 70% of the observed pili.

Altogether, our results support a model whereby PilY1, the four minor pilins, and the major pilin PilA form a priming complex that is essential for T4aP extension. This complex comprises the short stem and plug densities in the non-piliated T4aPM structure, and it is incorporated during assembly of the

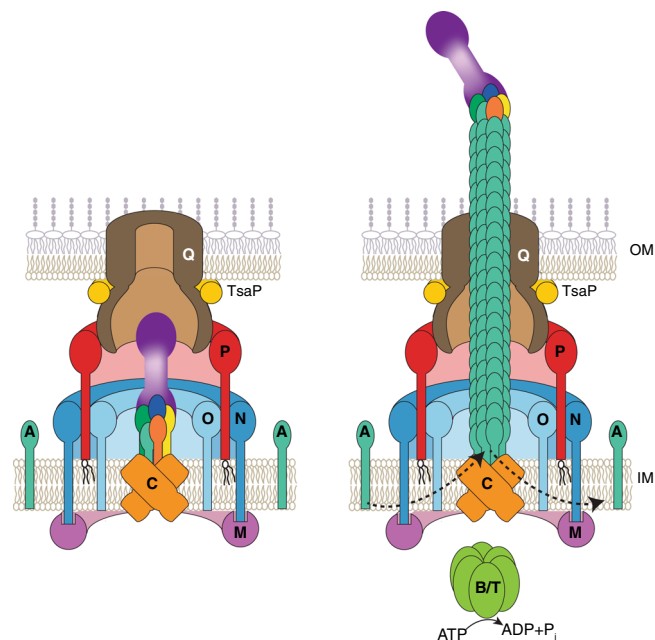

**Fig. 6 Three functions of the PilY1/minor pilin complex.** Left panel, architectural model of non-piliated T4aPM with ten core proteins and the priming complex composed of PilY1, four minor pilins, and one PilA subunit. Right panel, architectural model of piliated T4aPM with ten core proteins and the priming complex at the pilus tip; in the piliated T4aPM, PilB and PilT associate with PilC in a mutually exclusive manner during extension and retraction, respectively. For simplicity, PilB and PilT are not shown separately. Proteins labeled with single letters have the Pil prefix. Color code for PilY1 and minor pilins as in Fig. 4b.

T4aPM independently of the PilB extension ATPase (Fig. 6). In our model of the priming complex, PilY1.3 forms the plug at the entry to the PilQ secretin vestibule in the mid-periplasmic ring and the tip of the short stem while PilA and the minor pilins make up the rest of the short stem. The complex remains at the pilus tip during pilus extension from the cell and makes up the pilus tip (Fig. 6). In agreement with this model, previous immunogold labeling experiments indicated that PilY1 might be localized at the tip of T4aP in *N. gonorrhoeae*[20]. Because each T4aPM is likely to carry out many cycles of pilus extension and retraction, the consistent densities of the short stem and plug seen in WT non-piliated T4aPM by cryo-ET suggest that the priming complex remains intact after pilus retraction. We therefore suggest that the tip complex functions as a cork to prevent full retraction of the pilus into the IM, thereby terminating pilus retraction at the right place to ensure the formation of a T4aPM containing the priming complex in place for the next round of extension (Fig. 6). This stop–retraction mechanism may involve steric clashes between the PilY1.3 N-terminal vWFA domain (the plug density) with the mid-periplasmic ring and/or between the large globular domains of PilW3, FimU3, or the C-terminus of PilY1.3 with the lower periplasmic ring. Non-canonical arrangement of the N-terminal α-helices of the minor pilins in the priming complex caused by the packing of their varied globular domains and/or the binding with PilY1 is another possible mechanism.

Support for the idea that minor pilins are part of a priming complex also comes from the homologous T2SS. This system includes four minor pseudopilins and one major pseudopilin, which are homologs of minor pilins and the major pilin, respectively, and assemble to form a pseudopilus that translocates T2SS substrates across the OM[33]. Minor pseudopilins form a

complex[34,35] that has been suggested to prime pseudopilus formation[31] and to locate at the tip of the pseudopilus, which is mainly composed of the major pseudopilin[33]. The minor pseudopilin GspK is proposed to form the ultimate tip of the pseudopilus and interact with T2SS substrates[33]. Interestingly, GspK, similarly to PilX, lacks the conserved Glu5 residue in the mature protein[19] (Supplementary Fig. 1). In our model, PilX is at the tip of the major pilin/minor pilin complex and interacts with PilY1.3 suggesting that PilY1.3 is an analog of T2SS substrates.

Placing the variable N-terminal domain of PilY1 at the extreme pilus tip rationalizes how PilY1 might provide host cell specificity during infections, as suggested[16,18,20,36,37]. We speculate that the three separate PilY1/minor pilin gene clusters in *M. xanthus* enable the generation of priming complexes and T4aP tips with different properties, facilitating recognition of different substrates. Measurement of T4aP dynamics in *P. aeruginosa* has suggested that tip adhesion to a surface initiates retraction[38], supporting that the pilus tip has properties different from the rest of the pilus. Moreover, *P. aeruginosa* T4aP remain attached longer to a surface when tension is applied[38]. Consistently, PilY1 of *P. aeruginosa*, similarly to PilY1.3 of *M. xanthus* and PilY1 of *Legionella pneumophila*, contains an N-terminal vWFA domain that has been proposed to act as a mechanosensor involving a mechanism in which shear forces and/or surface contact induces conformational changes in the vWFA domain[14,27,39,40] making it tempting to speculate that PilY1 proteins, by analogy to the FimH tip adhesin of type 1 pili[41], might support surface adhesion via a catch-bond mechanism. Finally, we note that the tip complex in type I pili is assembled at the OM by the OM usher protein[41] while the tip complex of the needle in the type III secretion system is added to the needle tip after completion of the needle[42]. Thus bacteria have at least three different mechanisms for generating a tip complex on a filamentous cell surface structure.

## Methods

**Bacterial strains and growth media.** Strains and plasmids used in this study are listed in Supplementary Table 1. All *M. xanthus* strains are derivatives of DK1622[43]. In-frame deletion mutants were generated using *galK*-containing plasmids for double homologous recombination[44]. Plasmids for ectopic expression of genes in *M. xanthus* were integrated by site-specific recombination at the *attB* site and the relevant genes expressed from the *pilA* promoter. All plasmids were verified by sequencing. All strains were confirmed by PCR. Oligonucleotides are listed in Supplementary Table 2.

*M. xanthus* strains were grown at 32 °C in 1% CTT medium (1% casitone, 10 mM Tris-HCl pH 7.6, 1 mM KPO$_4$ pH 7.6, 8 mM MgSO$_4$) or on 1% CTT 1.5% agar plates supplemented with kanamycin (40 μg ml$^{-1}$) or oxytetracycline (10 μg ml$^{-1}$) when required[45]. *Escherichia coli* strains were grown in LB broth[46]. Plasmids were propagated in *E. coli* Mach1.

**T4aP-dependent motility assays.** Cells from exponentially growing *M. xanthus* cultures were harvested and resuspended in 1% CTT to a calculated density of 7 × 10$^9$ cells ml$^{-1}$. Five microliters were spotted on soft agar CTT plates (0.5% casitone, 10 mM Tris-HCl pH 7.6, 1 mM KPO$_4$ pH 7.6, 8 mM MgSO$_4$, 0.5% select agar (Invitrogen)) and incubated at 32 °C for 24 h. Colony edges were imaged using a Leica MZ75 stereomicroscope equipped with a Leica MC120 HD camera.

**T4aP shearing assays.** Pili were sheared of *M. xanthus* cells using a protocol based on the procedure of ref. [47]. Briefly, cells grown on 1% CTT and 1.5% agar plates for 2–3 days were gently scraped off the agar and resuspended in pili resuspension buffer (100 mM Tris-HCl pH 7.6, 150 mM NaCl) (1 ml per 60 mg cells). Cell suspensions were vortexed for 10 min at highest speed. Cells from a 100-μl aliquot were harvested, the pellet solved in 100 μl sodium dodecyl sulfate (SDS) lysis buffer (10% (v/v) glycerol, 50 mM Tris-HCl pH 6.8, 2 mM EDTA, 2% (w/v) SDS, 100 mM dithiothreitol, 0.01% bromphenol blue) and immediately denatured at 95 °C for 5 min. Total amount of protein was measured in cell extract samples using the detergent-compatible Bradford MX reagent (Expedion) and bovine serum albumin (BSA) standards in SDS lysis buffer. These samples were normalized to the same amount of protein and used to determine total cellular PilA amounts. The remaining suspension was centrifuged for 20 min at 13,000 × *g* at 4 °C. The supernatant was removed and centrifuged twice for 10 min at 13,000 × *g* at 4 °C to remove cell debris. T4aP in the cell-free supernatant were precipitated by adding 10× pili precipitation buffer (final concentrations: 100 mM MgCl$_2$, 2% PEG

6000, 100 mM Tris-HCl pH 7.6, 150 mM NaCl) for at least 2 h at 4 °C. The solution was centrifuged for 30 min at 13,000 × *g* at 4 °C, and the pellet was suspended in SDS lysis buffer (1 μl per mg vortexed cells). T4aP sheared and purified from the same amount of cells (normalized by protein determination, see above) were loaded and separated by SDS-polyacrylamide gel electrophoresis (PAGE) for immunoblot analysis.

**T4aP purification for SDS-PAGE and LFQ-MS analysis.** Pili were sheared from 60 mg *M. xanthus* cells using the same protocol as above with the following modification. Precipitation of T4aP was repeated twice. After the last precipitation, the pellet was resuspended in 60 μl pili resuspension buffer. In all, 50% were boiled in SDS lysis buffer, loaded on an SDS-PAGE, and stained with Coomassie protein stain (InstantBlue, Expedeon). The remaining 50% were precipitated with acetone for LFQ-MS analysis.

**Antibodies and immunoblot analysis.** For immunoblot analysis, exponentially growing *M. xanthus* cells were harvested and resuspended in SDS lysis buffer. Immunoblotting was done with rabbit polyclonal α-PilB, α-PilT[48], α-PilC, α-PilM, α-PilQ[24], α-PilN, α-PilO, α-PilP[23], α-TsaP[49], α-PilA, and α-LonD antibodies. Antibodies against PilA (MXAN_5783) were generated by Eurogentec against a His$_6$-Δ1-41-PilA protein purified from *E. coli* Rosetta 2 containing plasmid pMAT231 using affinity purification with a HiTrap Chelating column (GE Healthcare) loaded with CoCl$_2$. Antibodies against LonD (MXAN_3993) were generated by Eurogentec against LonD-His$_6$ purified from *E. coli* Rosetta 2(DE3) containing plasmid pSM30 using native Ni-NTA affinity purification. Polyclonal rabbit antibodies were used to detect FLAG-tagged proteins (Rockland; 600-401-383) and mCherry-tagged proteins (BioVision; 5993-100). Monoclonal mouse antibodies were used to detect GFP-tagged proteins (Roche; 11814460001) and CyaA-tagged proteins (Santa Cruz, sc-13582). As secondary antibodies, goat anti-rabbit immunoglobulin G peroxidase conjugate (Sigma-Aldrich, A8275) and sheep anti-mouse immunoglobulin G peroxidase conjugate (Amersham, NXA931) were used. Antibodies and conjugates were used in the following dilutions: 1:500 α-CyaA; 1:2000 α-PilM, α-PilN, α-PilO, α-TsaP, α-GFP, and anti-mouse peroxidase conjugate; 1:2500 α-FLAG and α -mCherry; 1:3000 α-PilB, α-PilT, and α-PilC; 1:5000 α-PilQ and α-PilA; 1:6000 α-LonD; 1:10,000 anti-rabbit peroxidase conjugate. Blots were developed using Luminata™ Western HRP substrate (Millipore). Unless otherwise noted, protein from 3 × 10$^8$ cells were loaded per lane.

**Fluorescence microscopy.** Five microliters of an exponentially growing *M. xanthus* culture was spotted on 1.5% agarose pads supplemented with TPM (10 mM Tris-HCl pH 7.6, 1 mM KPO$_4$ pH 7.6, 8 mM MgSO$_4$). Agar pads were incubated for 30 min at 32 °C in a humid chamber before microscopy. Cells were observed using a Leica DMI6000B microscope with a Hamamatsu Flash 4.0 camera. Images were recorded with Leica MM AF software package and processed with Metamorph (Molecular Devices). For each strain, at least 100 cells were analyzed.

**Bacterial Two-Hybrid.** BACTH experiments were performed according to the manufacturer's protocol (Euromedex). Briefly, DNA fragments encoding the sequences of the mature major and minor pilins as well as the AglQ control from *M. xanthus* were cloned into the vectors pKT25 and pUT18C to construct in-frame fusions at the C-terminal end of the T25 or T18 adenylate cyclase (CyaA) fragments. The AglR encoding sequence was cloned upstream of the N-terminal part of the T25 or T18 CyaA fragments using pKNT25 and pUT18 vectors. Subsequently, plasmids were transformed into *E. coli* BTH101 alone or in pairs. As a positive control, BTH101 co-transformed with the plasmids pKT25-zip and pUT18C-zip were used. In that control, the two leucine zippers dimerize giving rise to high β-galactosidase activity. cAMP production by reconstituted CyaA was qualitatively assessed by the formation of blue color as a read out for protein–protein interactions on LB agar supplemented with 40 μg ml$^{-1}$ 5-bromo-4-chloro-3-indolyl-β-D-galactopyranoside and 0.5 mM isopropyl-β-D-thiogalactopyranosid (IPTG) or quantitatively by measuring β-galactosidase activity using a 96-well microplate adapted protocol[50] with minor modifications. In brief, per transformation four randomly picked colonies were each transferred to 200 μl LB containing 50 μg ml$^{-1}$ ampicillin and 50 μg ml$^{-1}$ kanamycin per well and grown overnight at 32 °C. Cells (50 μl) were transferred to fresh 200 μl LB containing 50 μg ml$^{-1}$ ampicillin, 50 μg ml$^{-1}$ kanamycin, and 0.25 mM IPTG and grown for 4 h at 32 °C. Cells (50 μl) were diluted into 150 μl LB to measure optical density (OD$_{600 nm}$) in a microplate reader (Infinite M200, Tecan). Another aliquot of cells (50 μl) were permeabilized by transfer into 200 μl permeabilization solution (100 ml Z-buffer (60 mM Na$_2$HPO$_4$, 40 mM NaH$_2$PO$_4$, 10 mM KCl, 1 mM MgSO$_4$, 50 mM β-mercaptoethanol), 100 μl Lysonase (Merck, 71230), and 12.5 ml PopCulture reagent (Merck, 71092)) and further incubation for 15 min at room temperature (RT). Permeabilized cells (50 μl) were transferred into 150 μl Z-buffer. Per well, 40 μl ONPG (*O*-nitrophenyl-β-D-galactoside, 40 mg ml$^{-1}$) were added and OD$_{420nm}$ recording was started immediately at 28 °C every 2 min for 20 min in total in the microplate reader. The relative β-galactosidase activity in each of the 96 samples was calculated (OD$_{420nm}$ (t2) − OD$_{420 nm}$ (t1)/t2 − t1 (min)/OD$_{600nm}$). The t2 and t1 time points were chosen from the linear part of the ONPG conversion curve for each of the 96 samples.

To fractionate BTH101 cell extracts, cells were washed in phosphate-buffered saline (PBS) containing 2× protease inhibitor (see above), concentrated to $4 \times 10^{10}$ cells ml$^{-1}$ and sonicated. After centrifugation to remove cell debris and aggregated proteins (30 min, $10,000 \times g$ at 4 °C), the cleared cell lysates were fractionated in soluble and membrane fractions by ultracentrifugation (1 h, 4 °C, 28 psi, airfuge Beckman).

**Pull-down experiments**. Pull-down experiments were done using *M. xanthus* cells grown on 1% CTT and 1.5% agar plates. Cells (80 mg per replicate) were scraped from plates and washed in 1 ml PBS (137 mM NaCl, 2.7 mM KCl, 10 mM Na$_2$HPO$_4$, 1.8 mM KH$_2$PO$_4$, pH 7.5). Cells were suspended in 150 μl PBS, 150 μl detergent cocktail (8% CHAPS (3-[(3-cholamidopropyl)dimethylammonio]-1-propanesulfonate hydrate), 8% zwittergent 3–14 (3-[dimethyl(tetradecyl)azaniumyl]propane-1-sulfonate), 8% SLS (sodiumlauroylsarcosinate), 0.1% NP-40 (Nonidet P-40), 40% glycerol), and 33 μl protease inhibitor concentrate (1 tablet protease inhibitor completely dissolved in 1 ml PBS; Roche 11836145001) and lysis continued for 10 min at RT. The total volume was increased to 5 ml with PBS, up to 5 μl Benzonase (Merck, 101540001) were added, and the suspension was further incubated under rotation at 4 °C for 60 min. Cell debris was removed by centrifugation (15 min, $10,000 \times g$ at 4 °C). To the cleared supernatants, 15 μl of magnetic α-FLAG M2 beads (M8823, Sigma) were added for 1 h to capture FLAG-tagged proteins. To capture GFP-tagged proteins, 15 μl of GFP-Trap agarose (GTA-20, Chromotek) was added to the cleared supernatants and incubated for 1 h at 4 °C on a rotary shaker.

**Label-free quantitative mass spectrometry**. To determine the proteome of *M. xanthus* cells growing on 1% CTT 1.5% agar plates, 35 mg of cells were washed twice in PBS containing 2× protease inhibitor (see above). The final pellet was frozen in liquid nitrogen. Every pellet was suspended in 150 μl 2% SLS and 100 mM ammonium bicarbonate and heated for 60 min at 95 °C. The amount of extracted protein was measured using a BCA protein assay (Thermo Scientific). Seventy micrograms of protein was reduced using 5 mM Tris(2-carboxyethyl)phosphin (TCEP; Thermo Fischer Scientific) at 90 °C for 10 min and alkylated with 10 mM iodoacetamide (Sigma Aldrich) for 30 min at 25 °C in the dark. Proteins were precipitated using 400 μl ice-cold acetone followed by overnight incubation at −20 °C. Protein pellets were collected by centrifugation, and the remaining acetone was evaporated. Proteins were dissolved in 0.5% SLS solution using sonication. One microgram of trypsin (Promega) was added to 50 μg protein solution and digest was performed overnight at 30 °C. Following digest, peptides were acidified with trifluoroacetic acid (TFA; Thermo Fischer Scientific) and desalted using solid-phase extraction (SPE) on C18-Microspin columns (Harvard Apparatus). SPE columns were prepared by adding acetonitrile (ACN), followed by column equilibration with 0.1% TFA. Peptides were loaded on equilibrated Microspin columns and washed twice with 5% ACN/0.1% TFA. After peptide elution using 50% ACN/0.1% TFA, peptides were dried in a rotating concentrator (Thermo Fischer Scientific), reconstituted in 0.1% TFA, and subjected to liquid chromatography-mass spectrometry (LC-MS) analysis. LC-MS analysis was carried out on a Q-Exactive Plus instrument connected to an Ultimate 3000 RSLC nano with a Prowflow upgrade and a nanospray flex ion source (all Thermo Scientific). Peptide separation was performed on a reverse-phase HPLC column (75 μm × 42 cm) packed in-house with C18 resin (2.4 μm, Dr. Maisch). The following separating gradient was used: 98% solvent A (0.15% formic acid) and 2% solvent B (99.85 acetonitrile, 0.15% formic acid) to 35% solvent B over 120 min at a flow rate of 300 nl min$^{-1}$. The data acquisition mode was set to obtain one high-resolution MS scan at a resolution of 70,000 full width at half maximum (at $m/z$ 200) followed by MS/MS scans of the most intense ions. To increase the efficiency of MS/MS attempts, the charged state screening modus was enabled to exclude unassigned and singly charged ions. The dynamic exclusion duration was set to 30 s. The ion accumulation time was set to 50 ms for MS and 50 ms at 17,500 resolution for MS/MS. The automatic gain control was set to $3 \times 10^6$ for MS survey scans and $1 \times 10^5$ for MS/MS scans.

In pull-down experiments, beads were washed twice with 700 μl PBS–detergent wash buffer (3 ml PBS + 100 μl detergent cocktail) and five times with 700 μl of 100 mM ammonium bicarbonate (Sigma-Aldrich) to remove detergents. For elution, 100 μl elution buffer 1 (100 mM ammonium bicarbonate, 1 μg trypsin (Promega)) was added to each sample. After 30 min incubation at 30 °C, the supernatant containing the digested proteins was collected. Beads were washed twice with elution buffer 2 (100 mM ammonium bicarbonate, 5 mM TCEP) and added to the first elution fraction. Digestion was allowed to proceed overnight at 30 °C. Following digestion, the peptides were incubated with 10 mM iodoacetamide for 30 min at 25 °C in the dark. Prior to LC-MS analysis, peptides samples were desalted using C18 SPE. LC-MS analysis was performed as above, except that gradient length was reduced to 60 min.

For LC-MS analysis of sheared pili fractions, the dried pellets obtained by acetone precipitations were resuspended in 100 μl sodiumdeoxycholate buffer (2% sodiumdeoxycholate, 100 mM ammonium bicarbonate), heated for 15 min at 95 °C, reduced, and alkylated as described above. The detergent concentration was diluted to 0.5% and 30 μg pili was digested with 1 μg trypsin (Promega) overnight at 30 °C. Further SPE processing was carried out as described above. One microgram of total peptides were loaded onto the LC-MS system as described for the proteome samples. Settings for LC-MS analysis were also described earlier[51,52].

LFQ was performed using MaxQuant[53] and a *M. xanthus* protein database downloaded from UniProt. To calculate protein abundances and protein enrichment, iBAQ values[54] were calculated using MaxQuant. iBAQ values are calculated as the sum of all peptide intensities for a given protein divided by the number of theoretically MS observable peptides. Following MaxQuant analysis, the iBAQ values were normalized by the total iBAQ sum. The resulting MaxQuant output table was loaded into Perseus (v1.5.2.6)[55]. For calculation of enrichment factors in samples versus controls, only proteins with three or more peptides were considered; values for proteins not detected in the control were imputed using the imputation function from normal distribution implemented in Perseus in default settings (width, 0.3; down-shift, 1.8). Proteins with an absolute abundance difference of ≥2 (log2 ratio of ≥1) in the sample versus control and a $p$ value ≤0.05 using two-tailed Student's $t$ test were considered as enriched. Volcano plots were generated in Perseus. For purified T4aP, log2 enrichments were calculated based on normalized iBAQ values independently of the highly abundant PilA.

**Cryo-electron tomography**. Cryo-ET was done as described[3]. Briefly, *M. xanthus* cells were grown to an OD$_{550nm}$ of 0.8 in 1% CTT medium at 32 °C with 150 RPM of shaking. Ten-nanometer colloidal gold (Sigma-Aldrich, St. Louis, MO) pre-treated with BSA was added to the cells to serve as fiducial markers during tomogram reconstruction. Three microliters of the resulting sample was pipetted onto a freshly glow-discharged Quantifoil copper R2/2 200 EM grid (Quantifoil Micro Tools GmbH, Jena, Germany) and plunge-frozen in a liquid ethane propane mixture using an FEI Vitrobot mark-III (FEI Company, Hillsboro, OR). The frozen grids were imaged in an FEI Tecnai G2 Polara 300 keV FEG transmission electron microscope (FEI Company, Hillsboro, OR) equipped with a Gatan energy filter (Gatan, Pleasanton, CA) and a Gatan K2 Summit direct detector (Gatan, Pleasanton, CA). Energy-filtered tilt-series of images of cell poles were collected automatically from −60° to +60° at 1° intervals using the UCSF Tomography data collection software[56] with total dosage of 150 $e^-$ Å$^{-2}$, a defocus of −6 μm, and a pixel size of 3.9 Å. The images were aligned and contrast transfer function corrected using the IMOD software package[57]. SIRT reconstructions were then produced using the TOMO3D program[58]. T4aPM were located by visual inspection. Subtomogram averages with twofold symmetrization along the particle $Y$-axis were produced using the PEET program[59].

**Immunogold labeling and transmission electron microscopy**. To label T4aP attached to cells, exponentially growing *M. xanthus* cells in 1% CTT (OD$_{550 nm}$ = 0.5) were diluted fivefold in 1% CTT, placed in 6-well polystyrene plates (TC-plate F, Sarstedt 83.392), and allowed to adhere to the bottom. After 18 h, culture supernatants were removed and the adhered cells washed twice in PBS and once in 50 mM Na-Cacodylate buffer (stock 0.2 M pH7.4; EMS 11053). The adhered cells were overlaid with fixation solution (2% formaldehyde, 0.5% glutaraldehyde, in 50 mM Na-Cacodylate pH 7.4) and fixed for 3–5 h. Then the fixation solution was removed, and fixed cells were washed twice in PBS and incubated for 1 h in PBS containing 0.2% BSA (Aurion 900.099). The primary antibody (α-FLAG or α-mCherry, 1:200) was added and incubation was continued at 4 °C for 18 h. Then the fixed cells were washed twice in PBS and incubated for 1 h in 1 ml PBS plus 100 μl blocking reagent (Aurion 905.002). The secondary antibody was added (1:200, goat α-rabbit IgG gold, 10 nm colloidal gold, Sigma G7402). After 2 h, fixed cells were washed twice in PBS and carefully removed from the bottom of the plates. An aliquot of these cells was applied on copper grids (Plano, S162-3, 300 mesh). To label sheared T4aP, the primary antibody (α-FLAG or α-mCherry, 1:200) was added to aliquots of T4aP in pili resuspension buffer (s.a.) and incubation continued at 4 °C for 18 h. Then T4aP were precipitated with pili precipitation solution (s.a.). T4aP were washed in pili resuspension buffer and precipitated again. After resuspension of T4aP in resuspension buffer, the secondary antibody was added (1:200, goat α-rabbit IgG gold, 10 nm colloidal gold, Sigma G7402) for 2 h at 4 °C. T4aP were precipitated, washed, and resuspended before an aliquot of these T4aP solutions was applied on copper grids. After 30 min, grids were washed twice with water and negative staining was done with 2% uranyl acetate. Grids were inspected with a JEM-1400 electron microscope (JEOL) at 100 kV.

**Bioinformatics**. We previously identified *fimU*1–3, *pilV*1–3, *pilW*1–3, and the *pilX*1 gene[3] (Fig. 1b). Using the KEGG SSDB database[60], we identified a *pilX* gene in cluster_2. To identify PilY1-encoding gene(s) in *M. xanthus*, we used the PilY1 proteins from *P. aeruginosa* (PA_4554), *N. gonorrhoeae* (Z50180, Z49120), and *N. meningitidis* (NMV_0045 and NMV_2037) as queries in BlastP searches[61]. Using this approach, we identified two PilY1-encoding genes in the *M. xanthus* genome (*pilY1.2* and *pilY1.3*). Using the KEGG SSDB database[60], we additionally identified PilY1.1. Each of the three *pilY1* genes localize near the previously described three minor pilin gene clusters[3] (Fig. 1b). All proteins from cluster_1–_3 were analyzed using the program hmmscan on the HMMER web server[62]. These analyses revealed that MXAN_1366 in cluster_3 encodes a protein (PilX3), which similarly to PilX1 and PilX2 contained the conserved Pfam domain PF14341 (PilX N-term). Therefore, all three gene clusters encode a PilY1 protein and four minor pilins FimU, PilV, PilW, and PilX (Fig. 1b). Proteins were analyzed using SignalP v.4.1[63] to identify signal peptides with default gathering thresholds and the PHYRE2 protein fold recognition tool[64] to identify secondary structure elements and

conserved domains and to construct homology models. To identify domains in proteins, Pfam and HMM searches were performed using the program hmmscan on the HMMER web server[62]. The I-TASSER server was used for structure-based protein function annotation and for the generation of homology models[65]. Alignments were generated using T-Coffee[66] and the ClustalW output format[67]. The alignments were shaded using the BoxShade Server.

**Statistics and reproducibility**. Statistics were performed using a two-tailed Student's $t$ test for samples with equal variances using Perseus. Analysis of mean and standard deviation were done in Excel 2016. Data shown for T4aP-dependent motility, T4aP shear off experiments, immunogold experiments, immunoblot experiments, and fluorescence microscopy were obtained in at least two independent experiments with similar results. Localization patterns from fluorescence microscopic data are representative from $N > 100$ cells per strain. For qualitative and quantitative determination of protein–protein interactions using the BACTH, for any given combination four distinct clones were analyzed. For LFQ-MS analysis of proteome, three biological replicates were analyzed. For pull-down experiments, two biological replicates and two negative controls were used. Purification of T4aP pili and SDS-PAGE analysis were repeated several times with similar results. For LFQ-MS analysis on T4aP, samples from two biological replicates and two negative controls were analyzed. Multiple cryo-mograms were taken from each mutant. Multiple cryotomograms were taken to image pili tips.

**Reporting summary**. Further information on research design is available in the Nature Research Reporting Summary linked to this article.

## Data availability
The proteomics datasets are available via the PRIDE database (ProteomeXchange accession: PXD021163). We also used the following publicly available datasets: Uniprot (https://www.uniprot.org/) and KEGG (https://www.genome.jp/kegg/), including the KEGG SSDB database (https://www.kegg.jp/kegg/ssdb/). Source data are provided with this paper.

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

## Acknowledgements

We thank Ahmet Tekin for strains and plasmids and Magdalena Anna Świątek-Połatyńska for LonD antibodies. Cryo-ET work was done in the Beckman Institute Resource Center for Transmission Electron Microscopy at Caltech. This work was supported by the Deutsche Forschungsgemeinschaft (DFG) within the framework of the SFB987 "Microbial Diversity in Environmental Signal response" (to L.S.-A.), the Max Planck Society (to L.S.-A.), and NIH grant RO1 AI27401 (to G.J.J.).

## Author contributions

A.T.-L.: designed and conceived the study and performed most of the experiments. Y.-W.C.: performed the cryo-electron tomographic experiments and the modeling of the priming complex. T.G.: performed purification and analysis of pull-down samples and label-free mass spectrometry-based quantitative proteomics. S.L.: generated plasmids and strains and performed motility assays. M.H.: generated plasmids and strains and helped with transmission electron microscopy and bioinformatics. G.C.: helped with cryo-electron tomography data acquisition. G.J.J. and L.S.-A.: conceived the study, supervised research and provided funding. A.T.-L., Y.-W.C., T.G., G.J.J., and L.S.-A.: analyzed and interpreted data and wrote the manuscript. All authors approved the final manuscript.

## Funding

## Competing interests

The authors declare no competing interests.
