## [Peer Review File · Nature Communications]

Reviewers' comments:

Reviewer #1 (Remarks to the Author):

For the manuscript "PilY1 and minor pilins form a priming complex in the type IVa pilus machine and the pilus tip" the authors provide an impressive body of work. The manuscript is very carefully prepared and clearly written. The authors show convincingly that minor pilin/PilY1 clusters 1 and 3 are sufficient to produce type IV pili in *M. xanthus*, with cluster 3 being dominant; that the minor pilins and PilY1 are necessary for pilus production; that the pilus machinery (T4aPM) can assemble even in the absence of pilus assembly; that PilY1 localizes to the T4aPM; and that PilY1 requires the minor pilins but not the major pilin PilA to be expressed and to incorporate into the T4aPM. The final model of a minor pilin priming complex at the tip of the pilus is entirely plausible and has been proposed in several other publications; the suggestion that PilY1 is incorporated into the pilus tip is intriguing. However, the model and major claims of the manuscript, encapsulated in the title, are not well-supported by the data. Comments/critiques are listed below as the issues appear in the text.

General comment: Can some of the Supplemental Material be moved to the results section? It is problematic to have so much of the early results in the Supplemental section.

Lines 87-95. Clarify that there is a single major pilin and a single pilus machinery used by all three minor pilin clusters.

Line 94. Should say "all 13 major pilins are predicted to contain the N-terminal α -helix".

Fig. 1B. What is loaded here – whole cell culture? By "accumulation" do the authors mean "expression" Perhaps "levels" is a better term here and elsewhere?

Sup Fig. 1b. Provide a sequence alignment, at least for PilY1.3 and the *P. aeruginosa* PilY1, whose structure is available, and for a von Willebrand Factor A domain of known structure. Indicate the possible linker segment. Include in the labels the protein names as they are referred to in the text (and as shown in panel c).

Sup Fig. 1d. The pilin models are inherently unreliable due to the poor conservation in the C-terminal globular domains, consistent with the low C-scores. About all that can be gleaned by these models is that they have globular domains of an approximate volume, which is of little value.

BACTH, Fig. 2a, Sup Fig. 6a. While these assays have been used in other work to identify pilin:pilin interactions their utility is questionable. First, the mature pilins have the T18 or T25 adenylate cyclase fragment fused to their N-terminus. Was the pilin Type III signal sequence included as well? How are these fusion proteins localized to the inner membrane? The authors showed that T18-PilA is present in the insoluble fraction but this is not necessarily the membrane fraction – these proteins may just be insoluble and aggregating non-specifically. Second, assuming the pilins do get to the inner membrane, they possess a fragment of adenylate cyclase at their N-terminus, so they will be unable to assemble into pili or to form a functional priming complex, which requires that they be extracted out to the membrane and into the periplasm in varying degrees. Thus, any interactions identified would be between membrane-embedded pilins. Unless they all interact at once in the plane of the membrane, the results may not in fact define meaningful interactions within the priming complex. Were other pilus related and non-related inner membrane proteins examined as well to determine the specificity of these interactions? I would have more confidence in the data if the reciprocal interactions were consistent, but while T18-FimU interacts strongly with T25-PilW, -PilX and -PilA, only T18-PilA interacts with T25-FimU. Given this inconsistency, these experiments should be done in triplicate and quantified. Why does PilX interact with itself?

GFP and FLAG tags for PilY1.3 - specify that these are C-terminal tags.

Fig. 2d. Based on these and other data the authors propose that PilY1 forms a complex with the minor pilins at the tip of the pilus. Given this, wouldn't they expect that all minor pilins and the major pilin be immunoprecipitated by PilY1 given the likely tight interactions among the pilins? Certainly one would expect to see levels of all minor pilins and especially the major pilin over and above those of the T4bPM components. The fact that there are lower levels of the major pilin PilA than there are of the inner membrane PilO or the cytoplasmic ATPase motor PilT is concerning. What other proteins are pulled down with PilY1-FLAG? Where are the negative controls? How do we know that the pull-downs are specific and that many non-pilus-related inner/outer/periplasmic membrane proteins are not also pulled down?

Fig. 2d, 2h. Why do PilY1-GFP and PilY1-FLAG self-associate? Does this protein form a multimer? The authors state that FimU3, PilW3 and PilX3 are "highly enriched" in the PilY1-sfGFP pulldown but their values are not reported as significant. The value of PilY1-GFP self-IP is higher than that of PilX3. This should be addressed.

Line 166. Clarify that $\Delta 1\Delta 2\Delta 3pilY1$ is $\Delta 1\Delta 2$ cluster $\Delta 3pilY1$.

Fig. 3. Line 197-198 states that "PilA is not essential for PilY1 incorporation into the T4aPM but the minor pilins are." But the stem that the authors interpret as the minor pilin density is absent in the $\Delta pilA$ mutant (Fig. 3a column 2), suggesting that in the absence of PilA the minor pilin priming complex cannot form. So how do the minor pilins facilitate PilY1 incorporation into the T4aPM without forming a priming complex? The authors show that the minor pilins are required for PilY1 expression (Fig. 2f), and PilY1-GFP localizes to the poles in a $\Delta pilA$ mutant (Sup Fig. 7b). Does PilY1-GFP localize to the poles in a minor pilin mutant? If so, these data would support the interpretation of the plug-like density as being PilY1, but would also indicate that the minor pilins are not required for PilY1 incorporation into the T4aPM, and may not support the model shown in Fig. 3b.

Line 172. Define the $\Delta 9$ mutant.

The pseudoatomic model of the pilus tip is not really a pseudoatomic model since no atomic or near-atomic resolution structures are used. The authors use iTasser to generate pilin models, which are highly unreliable, they are filtered to the very low resolution of the sub-tomogram average of the pilus tip, are packed together based on questionable interaction data and then inserted into the 30 nm density of the pilus tip. Move any of those minor pilins to any location and you will likely see no difference in the model or the fit. The inclusion of PilY1 in this model, which is missing ~54% of the protein, adds a deeper level of uncertainty. Moreover, for the minor pilins to form a tip, their globular domains would interact much more intimately than they do in the body of the pilus, as there would be no N-terminal α -helices from pilins above to separate them. This appears to be what the authors have done, although it was not described, and does not appear to have been guided by any empirical data. The model is unconvincing.

Statements such as the "dimensions fit well" into the sub-tomogram averages, and "Remarkably, the dimensions of the kinked tip (average width ~4.5 nm, length ~12.5 nm) fit well to the PilY1.3 part of the priming complex (Fig. 4c)" are overblown as many shapes and sizes would fit into these densities.

Line 246. 10 μm seems very long. What is the average length of the *M. xanthus* pili?

Line 257. The pili used in this analysis would be quite short, much shorter than 10 μm . For instance, the pilus shown in Fig. 4c is ~ 0.15 μm . Could these just be broken pili? It is surprising that the kinked tip has not been reported previously for *M. xanthus* or other type IV pili that possess PilY1.

Line 293. What is the significance of the lack of conserved Glu5 for minor pilins proposed to be at the pilus tips (eg. PilX, GspK)?

Line 308-311. Not sure this point is meaningful. Differences in assembly of type IV pili and chaperone-usher pili are well established. The tip complex in type I pili represents the first pilins incorporated into the pilus, similar to what likely occurs for type IV pili, with or without incorporation of PilY1.

Reviewer #2 (Remarks to the Author):

In this paper Treuner-Lange and colleagues perform genetics and cryo-tomography experiments to establish the role of PilY and minor pilins in the T4aP system. This is an important open question and the authors do shed some partial new insights. Nevertheless, a number of conclusions are not evidence based and further experiments are required to confirm and consolidate the current presented experiments.

- The authors perform immune co-IP experiments where they use PilY1.3-FLAG to pull out T4aPM proteins. They do not detect PilA. I do not understand this as PilA should be pulled-out through its indirect interaction with minor pilins? Similar conclusions about indirect binding of proteins that are not pulled-out are made also in other cases, and later when designing a pseudo-atomic model they are used to 'rank' interaction strengths. I think this is not solid and interaction and affinity measurements with purified proteins should be made to establish the interaction network.
- Tomography in fig 3: while the general assignments made seem consistent with the data, the claim that the extra density in the plug corresponds to gfp is speculative. First the authors should demonstrate that the extra density is significant (is it the predominant density in a difference map of the average of mutants that are identical apart from the gfp tag?). If they cannot do this, I am afraid their further claims regarding the orientation of PilY1.3 are highly speculative and should be removed.
- One of the first conclusions, that the short stem and plug densities represent a priming complex for pilus formation, is not supported by the data. While the two phenotypes (lack of pilus and lack of the mentioned specific densities) are clearly related, there is no evidence that one (lack of pilus) is a direct consequence of the other (lack of short stem and plug). Alternative explanation to 'priming' should be discussed or at least acknowledged.
- The pseudo-atomic model reliability is quite weak, being based on co-IP and speculations on interaction strengths. It is not clear to me what value this model adds to the paper, and I suggest to remove it.
- The analysis of the tips does not provide evidence that PilY1.3 resides there. While the mass-spec analysis suggests that this is the case, direct evidence is lacking. I wonder whether immunogold labelling could be done for cryo-tomography? This should allow identification of the ends of pili.

Reviewer #3 (Remarks to the Author):

In the present manuscript Treuner-Lange et al. show a very extensive study of the role of 3 different pilY1 homologues and minor pilins clusters in the dynamics of Type IV pilus in the gram-negative bacterium *Myxococcus xanthus*. Combinatorial genetic mutants of the 3 different clusters

were combined with a wealth of techniques from biochemistry to structural cryo-EM to make a compelling case for the role of PilY1 and minor pilins in *Myxococcus xanthus* pili biogenesis. But starting with the title, not enough is done to explain the specificity of the *Myxococcus xanthus* model in terms of type IV pilus machinery and to contrast the present results with established results in other model systems. I will below summarize the points that I would wish to see addressed.

Major points:

- The dependence on pilQ to trigger the assembly of the type IV pilus machinery in an outside-in manner seems a specificity of *Myxococcus*. In *Neisseria gonorrhoeae* pilQ is not required for priming or elongation as a double pilQ+pilT mutant even shows polymerized pili accumulating in the periplasm (Wolfgang et al. EMBO Journal 19:6408-6418 (2000)).
- Still in *Neisseria gonorrhoeae*, using a similar FLAG tagging technique to the one presented in this manuscript it was shown that some minor pilins were confined to the periplasm (Imhaus et al. EMBO Journal 33:1767-1783 (2014)).
- Because of the previous points and the changing roles played by pilY1 across type IV pili systems I think it is capital that the specificity of the *Myxococcus xanthus* model would be extensively discussed and in particular that the title would be amended along those lines: "PilY1 and minor pilins form a priming complex in the type IV pilus machine and the pilus tip in the gram-negative bacterium *Myxococcus xanthus*"

Minor points:

- The presence of pili on the outside of the cells body is a complicated result of the ability or not of priming growth but also the actual ratio of elongation and retraction dynamics. Because of this, the term pili formation can be vague both referring to its nucleation and further extension and existence in an extended state. In particular in line 108, the term T4aP formation could be replaced by T4aP presence in sheared fraction or something of the kind.
- Line 194: when discussing the involvement of pilQ in incorporating pilY1 maybe once again make sure that this is contrasted with the fact that pilQ is not necessary for this step in other systems.
- The $\Delta 1\Delta 2$ cluster Δ pilT strain is well used as a control for pili presence in supplementary figure 3c, but that would be great if we could also have an image of the motility phenotype for the same strain added to this figure.
- In supplementary figure 3b, there are some interesting changes in the amount of sheared pili obtained for different mutants. In particular, the relative ratio seems to be inverted between $\Delta 1\Delta 2$ and $\Delta 2\Delta 3$ when we go from the entire cluster being deleted to pilX being deleting (second and fourth lane from the right). It seems to indicate a complex interaction between the different proteins that could be interesting to point out.

Reviewer #1 (Remarks to the Author):

For the manuscript “PilY1 and minor pilins form a priming complex in the type IVa pilus machine and the pilus tip” the authors provide an impressive body of work. The manuscript is very carefully prepared and clearly written. The authors show convincingly that minor pilin/PilY1 clusters 1 and 3 are sufficient to produce type IV pili in *M. xanthus*, with cluster 3 being dominant; that the minor pilins and PilY1 are necessary for pilus production; that the pilus machinery (T4aPM) can assemble even in the absence of pilus assembly; that PilY1 localizes to the T4aPM; and that PilY1 requires the minor pilins but not the major pilin PilA to be expressed and to incorporate into the T4aPM. The final model of a minor pilin priming complex at the tip of the pilus is entirely plausible and has been proposed in several other publications; the suggestion that PilY1 is incorporated into the pilus tip is intriguing. However, the model and major claims of the manuscript, encapsulated in the title, are not well-supported by the data. Comments/critiques are listed below as the issues appear in the text.

Response: Thank you very much for the encouraging comments. Inspired by the constructive criticism, we have now included several new experiments and repeated other experiments. Altogether, these revisions strengthen our conclusions.

General comment: Can some of the Supplemental Material be moved to the results section? It is problematic to have so much of the early results in the Supplemental section.

Response: We followed the advice of the reviewer and moved the figure with the three gene clusters (previously Fig. S1a) and the characterization of mutants with one or more clusters, minor pilins or PilY1 proteins missing (Previously Fig. S3a-c) to Fig. 1 in the main text.

Lines 87-95. Clarify that there is a single major pilin and a single pilus machinery used by all three minor pilin clusters.

Response: We followed the advice of the reviewer and added (line 111-113) that “Because *M. xanthus* has a single major pilin, these analyses suggest that the 12 minor pilins and three PilY1 proteins function together with one major pilin and a single T4aPM in *M. xanthus*”.

Line 94. Should say “all 13 major pilins are predicted to contain the N-terminal α -helix”.

Response: Thanks and done (line 100).

Fig. 1B. What is loaded here – whole cell culture? By “accumulation” do the authors mean “expression” Perhaps “levels” is a better term here and elsewhere?

Response: Fig. 1b is now Fig. 1f. We changed “accumulation” to “accumulation levels” throughout.

Sup Fig. 1b. Provide a sequence alignment, at least for PilY1.3 and the *P. aeruginosa* PilY1, whose structure is available, and for a von Willebrand Factor A domain of known structure. Indicate the possible linker segment. Include in the labels the protein names as they are referred to in the text (and as shown in panel c).

Response: The PilY1 proteins are now described in details in Fig. S2 and S3. We have included in Fig. S2b a sequence alignment of the C-terminal conserved PilY1 domain of the three *M. xanthus* PilY1 proteins together with the C-terminal conserved PilY1 domain of the two *N. gonorrhoeae* PilY1 proteins and that of the PilY1 protein of *P. aeruginosa* (for which the structure is available). In Fig. S3a, we have included an alignment of the von Willebrand Factor A (vWFA) domain of PilY.3 of *M. xanthus* and of PilY1 of *P. aeruginosa* together with the vWFA domain of the *Catenulispora acidiphila* protein (Caci_2163) for which the structure (4FX5) is available. We have also made sure that we use the same nomenclature for the aligned proteins throughout. In Fig. S2a, we have indicated the two domains of PilY1.3 that make up 46% of the protein.

Sup Fig. 1d. The pilin models are inherently unreliable due to the poor conservation in the C-terminal globular domains, consistent with the low C-scores. About all that can be gleaned by these models is that they have globular domains of an approximate volume, which is of little value.

Response: These models are now included in Fig. S11.

We absolutely agree with the reviewer that the homology models of the globular domains of the pilins may not be correct in all details and that their value is to estimate the approximate volume these proteins could occupy in space. We believe that they add significant value to the manuscript by supporting that the densities of short stem and plug together are sufficient to accommodate at least one copy of each of the priming complex components (PilA, four minor pilins, and PilY1.3). This serves as a valid structural examination to support our conclusions drawn in the manuscript title. We note that while we have tried our best to arrange the pilin subunits in the model in a fashion that agrees with the results of our comprehensive biochemistry assay for protein-protein interactions as well as other available knowledge, we did not put emphasis on the arrangement nor any atomic details of the pilin subunits other than just filtered the overall model to 3 nm resolution and used its size to test and support the assignment of corresponding subtomogram average densities. However, we absolutely understand the reviewer's concern regarding that readers might overly interpret the information carried by the structural model. We have therefore toned down our language in the text to reflect the role of the model in our line of arguments (line 290-318). Also, we now refer to the model of the priming complex as a "hypothetical structural model" (and not as a pseudo-atomic model). We moved the stepping-stones in the generation of the hypothetical structural model of the priming complex from a main text figure to Fig. S11a,b and now only include the model filtered to 3 nm resolution together with a component map of the T4aPM in Fig. 4b. Moreover, in Fig. S11a we clearly include the C-scores and we also write in the legend to Fig. S11a that "C-scores are expressed in numbers from -5 to +2, with +2 indicating high confidence".

BACTH, Fig. 2a, Sup Fig. 6a. While these assays have been used in other work to identify pilin:pilin interactions their utility is questionable. First, the mature pilins have the T18 or T25 adenylate cyclase fragment fused to their N-terminus. Was the pilin Type III signal sequence included as well? How are these fusion proteins localized to the inner membrane? The authors showed that T18-PilA is present in the insoluble fraction but this is not necessarily the membrane fraction – these proteins may just be insoluble and aggregating non-specifically. Second, assuming the pilins do get to the inner membrane, they possess a fragment of adenylate cyclase at their N-terminus, so they will be unable to assemble into pili or to form a functional priming complex, which requires that they be extracted out to the membrane and into the periplasm in varying degrees. Thus, any interactions identified would be between membrane-embedded pilins. Unless they all interact at once in the plane of the membrane, the results may not in fact define meaningful interactions within the priming complex. Were other pilus related and non-related inner membrane proteins examined as well to determine the specificity of these interactions? I would have more confidence in the data if the reciprocal interactions were consistent, but while T18-FimU interacts strongly with T25-PilW, -PilX and – PilA, only T18-PilA interacts with T25-FimU. Given this inconsistency, these experiments should be done in triplicate and quantified. Why does PilX interact with itself?

Response: The BACTH system has previously been used successfully to detect interactions between minor and major pilins as well as between minor and major pseudopilins of the type II secretion system. We have included 7 references to these studies in line 151. As was also the case in these prior studies, our constructs do not include the type III signal peptide and the T18 and T25 fragments were fused to the N-terminus of the mature pilins (line 151-152) and Fig 2a). So, we are using an established methodology to test for interactions between minor and major

pilins. Inspired by the comments of the reviewer, we included two additional control experiments to test whether these interactions are specific:

1. As described in line 155-159 “In control experiments for specificity, we observed that neither PilA nor minor pilins fusions interacted with the bitopic IM protein AglQ, a MotA homolog, fused to the C-terminus of the T18 or T25 fragments (Fig. 2a) while both AglQ constructs interacted with its IM partner protein AglR, a MotB homolog, as previously shown (ref32) (Supplementary Fig. 7a)”.

2. As described in line 159-160 “Moreover, in fractionation experiments, we observed that all six T18 fusions, and therefore likely also the T25 fusions, were enriched in the membrane fraction (Fig. 2b)”. In these fractionation experiments, the *E. coli* cells were lysed by sonication, and then as described in line 502-505 centrifuged “...to remove cell debris and aggregated proteins (30 min, 10,000 *g* at 4°C), the cleared cell lysates were fractionated in soluble and membrane fractions by ultracentrifugation (1 h, 4°C, 28psi, airfuge Beckman)”.

3. Finally, we observed in line 163-165 that “a T18 fusion to a PilA variant lacking the N-terminal segment of the N-terminal α -helix that anchors the protein in the IM did not interact with the minor pilins and was not enriched in the membrane fraction (Supplementary Fig. 7b,c)”.

We also specifically included in the legend to Fig. 2a that “For every tested interaction pair, four clones were tested with similar results in two biological replicates”. Finally, we included measurements to determine specific β -galactosidase activities (Fig. 2a).

Altogether, these observations demonstrate that the fusion proteins localize to the inner membrane, that the interactions detected occur between proteins in the membrane, and that these interactions are specific. And, we write in line 165-167 “Altogether, these results demonstrate that mature PilA and minor pilins interact and are in agreement with the formation of a complex containing all five pilins”.

Concerning the lack of reciprocal interactions, this is commonly observed in BACTH analyses. We find it interesting that PilX self-interacts and mention this self-interaction in the text (line 153) but do not further discuss this observation.

In the revised manuscript, we included several additional experiments in which we analyze a PilW3-sfGFP fusion (line 182-212). In pull-down experiments with this fusion (which is incorporated into the T4aPM but is not active), PilX3 and PilY1.3 (in addition to PilO and PilB) are enriched (Fig. 3h), thus, further corroborating that the minor pilins interact.

GFP and FLAG tags for PilY1.3 - specify that these are C-terminal tags.

Response: Done in line 173.

Fig. 2d. Based on these and other data the authors propose that PilY1 forms a complex with the minor pilins at the tip of the pilus. Given this, wouldn't they expect that all minor pilins and the major pilin be immunoprecipitated by PilY1 given the likely tight interactions among the pilins? Certainly one would expect to see levels of all minor pilins and especially the major pilin over and above those of the T4bPM components. The fact that there are lower levels of the major pilin PilA than there are of the inner membrane PilO or the cytoplasmic ATPase motor PilT is concerning. What other proteins are pulled down with PilY1-FLAG? Where are the negative controls? How do we know that the pull-downs are specific and that many non-pilus-related inner/outer/periplasmic membrane proteins are not also pulled down?

Response: We apologize very much for this confusion! We repeated all the pull-down experiments with PilY1.3-FLAG (Fig. 3c) and PilY1.3-sfGFP (Fig. 3g) and included two new pull-downs with PilW3-sfGFP (Fig. 3g) and PilY1.3 Δ vWFA-FLAG (Fig. S9). In all four experiments,

we used newly constructed strains deleted for cluster_1 and cluster_2 to have more clean systems. All pull-down experiments were done with two biological replicates and two negative controls allowing us to do statistical test for significant differences.

In Fig. S8a-d, we have included volcano plots of proteins enriched in the pull-down experiments. Moreover, these proteins are listed in the Source Tables. In the description of these experiments, we write throughout the text “Among proteins known to be related to T4aP formation and function” to describe enriched proteins.

In Fig. S8e, we included Venn diagram comparisons between relevant pull-downs as a test for specificity and write in the text (line 207-211) “Moreover, pairwise comparisons of enriched proteins in the PilY1.3-FLAG, PilY1.3-sfGFP and PilW3-sfGFP pull-down experiments, revealed only two additional proteins that were enriched in all three experiments (Supplementary Fig. 8e), both of which are predicted to be cytoplasmic proteins and none of which have been associated with T4aP formation and function”.

Finally, regarding PilA: PilA is indeed detected in all the pull-down experiments with FLAG-tagged or sfGFP-tagged proteins. However, PilA binds non-specifically to the α -FLAG matrix and α -GFP matrix used in the pull-down experiments. Moreover, our label-free quantitative mass spectrometry (LFQ-MS) on whole cell extracts revealed that PilA is one of the most abundant proteins in *M. xanthus* (line 136-137). Because PilA is so highly abundant, this precludes its analysis by this method. Accordingly, we write in the text to all the pull-down experiments that “PilA is highly abundant and binds non-specifically to the α -FLAG matrix (α -GFP matrix), precluding its analysis by this method”.

To the specific questions:

In the PilY1.3-FLAG (which is active) pull-down, we observed enrichment of the minor pilins PilX3, PilW3, FimU3 and the core protein PilO.

In the experiments with PilY1.3-sfGFP (which is not active) pull-down, we observed enrichment of the same three minor pilins and core protein TsaP.

In the newly added PilW3-sfGFP (which is not active) pull-down, we observe enrichment of PilY1.3 and PilX3.

In the newly added PilY1.3 Δ vWFA-FLAG (which is not active) pull-down, we observed enrichment of the same three minor pilins and PilO.

Thus, we find that PilY1.3 pulls down 3 of the cluster_3 minor pilins and PilW3 pulls-down PilY1.3.

Finally, in a new set of experiments included in Fig. 3def (and described in line 182-200), we find that PilW3-sfGFP stability and incorporation into the T4aPM depends on PilY1.3 and PilY1.3 stability depends on minor pilins (specifically PilX) and its incorporation into the T4aPM depends on PilW3, FimU3 and PilV3.

Based on all these experiments we write that (line 213-217) “Together, these observations support that (1) PilY1.3 and PilW3, and therefore, more generally cluster_3 minor pilins are incorporated into the T4aPM, (2) minor pilins and PilY1.3 interact within the T4aPM, and (3) their incorporation into the T4aPM is mutually dependent but independent of PilA. Moreover, the bulky PilY1.3-sfGFP and PilW3-sfGFP proteins are incorporated into the T4aPM, but somehow jam the machine.

The negative control in the pull-downs is GFP-FLAG, i.e. all results from the pull-downs are shown as enrichment factors (sample over negative control).

Fig. 2d, 2h. Why do PilY1-GFP and PilY1-FLAG self-associate? Does this protein form a multimer? The authors state that FimU3, PilW3 and PilX3 are “highly enriched” in the PilY1-sfGFP pulldown but their values are not reported as significant. The value of PilY1-GFP self-IP is higher than that of PilX3. This should be addressed.

Response: The former 2d and 2h are now Fig. 3c and 3g. In these experiments we use PilY1-GFP and PilY1-FLAG (and GFP-FLAG as negative control) as baits in the pull-downs and, therefore, they are enriched in these experiments. The enrichment does not reflect self-interaction.

In the original submission, the data for the pull-down experiments with PilY1.3-sfGFP were not reported with significance statements because we only had one negative control. As described above, we have repeated all the pull-down experiments with two biological samples and two negative controls. Therefore, in the new figures we have now included significance statements.

Line 166. Clarify that $\Delta 1\Delta 2\Delta 3pilY1$ is $\Delta 1\Delta 2cluster \Delta 3pilY1$.

Response: We apologize for not being clear and rewrote to “...imaged a $\Delta 1\Delta 2\Delta 3pilY1$ mutant that lacks all three PilY1 proteins” (line 220).

Fig. 3. Line 197-198 states that “PilA is not essential for PilY1 incorporation into the T4aPM but the minor pilins are.” But the stem that the authors interpret as the minor pilin density is absent in the $\Delta pilA$ mutant (Fig. 3a column 2), suggesting that in the absence of PilA the minor pilin priming complex cannot form. So how do the minor pilins facilitate PilY1 incorporation into the T4aPM without forming a priming complex? The authors show that the minor pilins are required for PilY1 expression (Fig. 2f), and PilY1-GFP localizes to the poles in a $\Delta pilA$ mutant (Sup Fig. 7b). Does PilY1-GFP localize to the poles in a minor pilin mutant? If so, these data would support the interpretation of the plug-like density as being PilY1, but would also indicate that the minor pilins are not required for PilY1 incorporation into the T4aPM, and may not support the model shown in Fig. 3b.

Response: This was a writing mistake in the original submission and we apologize for this. In the revised manuscript, we now clearly describe the logic in our assignment of densities to specific proteins. Based on the tomography of the $\Delta 1\Delta 2\Delta 3pilY1$ mutant, the $\Delta pilA$ mutant and the $\Delta 9$ minor pilin mutant together with the fluorescence microscopy of PilY1.3-sfGFP and PilW3-sfGFP, we write in line 231-234 “... these data also support that the lower part of the short stem is accounted for by PilA. Because minor pilins and PilY1.3 are mutually dependent for incorporation into the T4aPM, the assignment of specific arrangement among these proteins was less clear”.

To disentangle the densities accounted for by the minor pilins and PilY1, we did three experiments:

1. We imaged by cryo-ET, the strain containing PilY1.3-sfGFP as the only PilY1 protein and write based on this experiment in line 249-252 “..... these data support that PilY1.3’s conserved C-terminal PilY1 domain is located at the tip of the short stem in the non-piliated T4aPM structure, and the less conserved N-terminal domain accounts for the plug density at the entry to the PilQ cavity”.

2. In two new experiments we analyzed a strain containing a FLAG-tagged PilY1.3 variant deleted for its N-terminal vWFA-domain (PilY1.3 Δ vWFA-FLAG) as the only PilY1 protein. Based on the new pull-down experiments with this protein (Fig. S9b) and the new cryo-ET experiments (Fig. 4a, column 7), we write line 264-267 “Combined these observations support the model whereby the C-terminal PilY1 domain is part of the short stem and the vWFA domain of PilY1.3 account for the plug density at the PilQ vestibule entry (Fig. 4b, left panel)”.

Line 172. Define the $\Delta 9$ mutant.

Response: We apologize for not being clear. We rewrote to “..in a $\Delta 1\Delta 2\Delta 3fimUpilVpilW$ mutant (henceforth, $\Delta 9$ minor pilin mutant)” (line 226).

The pseudoatomic model of the pilus tip is not really a pseudoatomic model since no atomic or near-atomic resolution structures are used. The authors use iTasser to generate pilin models, which are highly unreliable, they are filtered to the very low resolution of the sub-tomogram average of the pilus tip, are packed together based on questionable interaction data and then inserted into the 30 nm density of the pilus tip. Move any of those minor pilins to any location and you will likely see no difference in the model or the fit. The inclusion of PilY1 in this model, which is missing ~54% of the protein, adds a deeper level of uncertainty. Moreover, for the minor pilins to form a tip, their globular domains would interact much more intimately than they do in the body of the pilus, as there would be no N-terminal α -helices from pilins above to separate them. This appears to be what the authors have done, although it was not described, and does not appear to have been guided by any empirical data. The model is unconvincing.

Response: We agree with the reviewer that the term ‘pseudoatomic model’ is inappropriate to be used here. We have therefore removed it from the text and used ‘hypothetical structural model’ instead. We know that computational protein structure prediction still has a long way to go, especially when there is a lack of available atomic structures from highly homologous proteins as a template. So, we absolutely understand the reviewer’s concern about the unreliability of the predicted structures of the minor pilins. We note that the models were generated by the I-TASSER server, which is a sophisticated system that takes the results of a protein sequence searching against available structural knowledge and uses multiple threads to plug all the best-predicted structural fragments into an ab initio simulation, which tries to minimize the energy of the entire model by sampling many possible 3D conformations. While the method has demonstrated ample success on structure prediction of various proteins as shown on the I-TASSER server, we are as cautious as the reviewer is and did not want to rely on any of the atomic details other than just the size of the models to conduct the analysis in this study. We believe the size of the predicted models should be reliable, as globular protein domains (e.g. minor pilin heads) have a similar density. We have also toned down and corrected our language in the text to make it clear that the purpose of the structural model of the priming complex is to test whether the short stem and plug densities are feasible to accommodate all the proposed protein components (line 290-318).

Statements such as the “dimensions fit well” into the sub-tomogram averages, and “Remarkably, the dimensions of the kinked tip (average width ~4.5 nm, length ~12.5 nm) fit well to the PilY1.3 part of the priming complex (Fig. 4c)” are overblown as many shapes and sizes would fit into these densities.

Response: Point well taken and we changed the text as suggested by the reviewer (line 290-318 and line 352-354)

Line 246. 10 μm seems very long. What is the average length of the *M. xanthus* pili?

Response: In Fig. 5b, for simplicity, we are normalizing the absolute iBAQ values for PilY.3 and cluster_3 minor pilins to an iBAQ value of 10,000 for PilA, which corresponds to a 10- μm long pilus. We have now included that T4aP in *M. xanthus* are typically 2-6 μm in length (line 335).

Line 257. The pili used in this analysis would be quite short, much shorter than 10 μm . For instance, the pilus shown in Fig. 4c is ~ 0.15 μm . Could these just be broken pili?

Response: In these analyses (described in line 346-367), we searched for T4aP in the cryo-tomograms that could be tracked all the way from the T4aPM in the cell envelope to the pilus tip. We did this to minimize the chances that we would look at pili that had been broken off from the cells (and where we would not know whether a pilus end would represent the beginning or the end of a pilus). Because the high magnification required for high-resolution cryo-ET results in a limited field-of-view (in this case is about 1 μm^2), only shorter pili could be followed from a cell to the tip. We have now included this additional information in line 348-350. We also note in line 362-367 “At this point, we could not rule out the possibility that some of the pilus tips observed were from snapped pili due to their surface binding and retraction. However, since the cells used for these analyses were cultured in suspension (Methods) with limited chance for surface binding before being frozen directly for cryo-ET imaging, we speculate that if there are broken pili, they would only account for a minority of pili tips in the whole population and not represent the kinked structure in 70% of observed pili”.

It is surprising that the kinked tip has not been reported previously for *M. xanthus* or other type IV pili that possess PilY1.

Response: We were also surprised when we noticed the kinked T4aP tip in cryo-tomograms, since we have previously looked at many, many T4aP (either sheared and purified from cells or attached to cells) by negative stain TEM and never seen a clear tip structure. We believe it is because that through plunge-freezing, the structure was better preserved in a frozen-hydrated state. Cutting-edge cryo-ET imaging also played an important role to reveal that unprecedented detail on individual pilus tips. We believe this result will stimulate other researchers to use the same approach to identify tip structures in other systems.

Line 293. What is the significance of the lack of conserved Glu5 for minor pilins proposed to be at the pilus tips (eg. PilX, GspK)?

Response: We propose that PilX is at the tip of the major pilin/minor pilin complex and interacts with PilY1.3 (line 393). We find it interesting that GspK, which has been proposed to form the ultimate tip of the pseudopilus and to interact with T2SS substrates also lacks the Glu5 residue. In major pilins the Glu 5 residue is involved in the interactions between the N-terminal α -helix in the pilus structure. The observation that PilX does not have this residue is interesting but we believe that it is premature to discuss this in details in this manuscript.

Line 308-311. Not sure this point is meaningful. Differences in assembly of type IV pili and chaperone-usher pili are well established. The tip complex in type I pili represents the first pilins incorporated into the pilus, similar to what likely occurs for type IV pili, with or without incorporation of PilY1.

Response: We think this comparison is informative and promotes the thinking of similarities and differences among important bacterial filamentous appendages and would really like to keep it in the manuscript.

Reviewer #2 (Remarks to the Author):

In this paper Treuner-Lange and colleagues perform genetics and cryo-tomography experiments to establish the role of PilY and minor pilins in the T4aP system. This is an important open question and the authors do shed some partial new insights. Nevertheless, a number of conclusions are not evidence based and further experiments are required to confirm and consolidate the current presented experiments.

Response: Thank you very much for the encouraging comments. Inspired by the constructive criticism, we have now included several new experiments and repeated other experiments. Altogether, these revisions strengthen our conclusions.

- The authors perform immune co-IP experiments where they use PilY1.3-FLAG to pull out T4aPM proteins. They do not detect PilA. I do not understand this as PilA should be pulled-out through its indirect interaction with minor pilins? Similar conclusions about indirect binding of proteins that are not pulled-out are made also in other cases, and later when designing a pseudo-atomic model they are used to 'rank' interaction strengths. I think this is not solid and interaction and affinity measurements with purified proteins should be made to establish the interaction network.

Response: We apologize for not being more clear in the original manuscript.

We repeated all the pull-down experiments with PilY1.3-FLAG (Fig. 3c) and PilY1.3-sfGFP (Fig. 3g) and included two new pull-downs with PilW3-sfGFP (Fig. 3h) and PilY1.3 Δ vWFA-FLAG (Fig. S9). In all four pull-down experiments, we used newly constructed strains deleted for cluster_1 and cluster_2 to have more clean systems. All pull-down experiments were done with two biological replicates and two negative controls allowing us to do statistical test for significant differences. In Fig. S8, we have included volcano plots of proteins enriched in the pull-downs. Moreover, these proteins are listed in the Source Tables. Also, in Fig. S8, we included Venn diagram comparisons between relevant pull-downs as a test for specificity.

Regarding PilA: PilA is indeed detected in all the pull-down experiments with FLAG-tagged or sfGFP-tagged proteins. However, PilA binds non-specifically to the α -FLAG matrix and α -GFP matrix used in the pull-down experiments. Moreover, our label-free quantitative mass spectrometry (LFQ-MS) on whole cell extracts revealed that PilA is one of the most abundant proteins in *M. xanthus* (line 136-137). Because PilA is so highly abundant, this precludes its analysis by this method. Accordingly, we write in the text to all the pull-down experiments that "PilA is highly abundant and binds non-specifically to the α -FLAG matrix (α -GFP matrix), precluding its analysis by this method". Altogether, we find in the pull-downs that PilY1 pulls-down minor pilins. In new experiments we find that the minor pilin-sfGFP fusion PilW3-sfGFP pulls-down PilY1.3. Based on the BACTH experiments, we know that PilA interacts with the minor pilins.

Regarding the pseudo-atomic model: We have toned down our language in the description of this model and now refer it as a "hypothetical structural model". With the new pull-down experiments with PilW3-sfGFP and PilY1.3 Δ vWFA-FLAG as well as the new experiments in which we look at PilY1.3-sfGFP stability in the absence of individual cluster_3 minor pilins and PilW3-sfGFP stability and incorporation into the T4aPM (Fig. 3def), we have additional support for ordering of the minor pilins in the priming complex. We would like to add that the "hypothetical structural model" is a model and we are only using it to probe whether a complex consisting of 1 PilA, 4 minor pilins and 1 PilY1 protein could fit into the short stem and plug densities. We write in line 307-310 "The dimensions of the generated structural envelopes of the complex composed of one PilA/four minor pilins/one PilY1 domain and the N-terminal vWFA domain of PilY1.3 fit into and do not exceed the short stem and plug, respectively in the subtomogram average of WT T4aPM". Moreover, we write (line 314-318) "This model successfully confirmed that the short stem and plug densities are feasible for accommodating at least one of each of the assigned protein components; however, the exact structure of the globular domains of the pilins, their precise order and packing, as well as the stoichiometry of the involved proteins remain to be revealed by future studies".

Finally, we would like to add that it is a great idea to study the interactions and do affinity measurements with purified proteins to establish the interaction network. But these measurements are currently hampered by the difficulties purifying full-length pilins. In fact, to our knowledge there are no published reports on purified full-length pilins. We are momentarily trying to purify the priming complex. Without going into details here, it is already clear that this is

not straight forward. So, we consider these experiments well beyond the scope of the current manuscript.

- Tomography in fig 3: while the general assignments made seem consistent with the data, the claim that the extra density in the plug corresponds to gfp is speculative. First the authors should demonstrate that the extra density is significant (is it the predominant density in a difference map of the average of mutants that are identical apart from the gfp tag?). If they cannot do this, I am afraid their further claims regarding the orientation of PilY1.3 are highly speculative and should be removed.

Response: A subtomogram average is a result of retaining consistent densities and removing inconsistent densities among all molecules used in the average, so all densities revealed are considered significant. When we compared the two subtomogram averages of *M. xanthus* T4aPM with and without a sfGFP-tag on PilY1.3, the only major difference is at the tip of the short stem with a corresponding size of a sfGFP (~4nm in diameter; Fig. 4a column 5-6). We therefore argue that the difference is significant and the assignment of the sfGFP to the location for orienting PilY1.3 is not highly speculative. The same workflow has previously successfully allowed us to map other components (PilP and PilO) in the same T4aPM (REF 3). However, we understand the reviewer's concern and have therefore performed a new set of experiments to strengthen our argument. We first made a new strain where the N-terminal vWFA domain was truncated from PilY1.3 (PilY1.3 Δ vWFA-FLAG). We then did pull-down experiments (Fig. S9) on this strain and find that the PilY1.3 Δ vWFA-FLAG protein pulled-down exactly the same proteins (three minor pilins and PilO) as PilY1.3-FLAG did, demonstrating that it is the C-terminal conserved PilY1 domain interacting with the minor pilins and PilO. Since PilO is in the lower-periplasmic ring and minor pilins are in the short stem, the result agrees with our assignment that the PilY1 domain is on the tip of short stem. Finally, we imaged this new strain by cryo-ET and produced the subtomogram average of its T4aPM (Fig. 4a, column 7). The subtomogram average clearly lacked the plug density, while the short stem was reduced in intensity. The observation again agrees with our assignment of the vWFA domain being in the plug density.

- One of the first conclusions, that the short stem and plug densities represent a priming complex for pilus formation, is not supported by the data. While the two phenotypes (lack of pilus and lack of the mentioned specific densities) are clearly related, there is no evidence that one (lack of pilus) is a direct consequence of the other (lack of short stem and plug). Alternative explanation to 'priming' should be discussed or at least acknowledged.

Response: We observed that (1) minor pilins and PilY1 extensively interact with one another as well as with other T4aPM core proteins, (2) minor/major pilins and PilY1 together form the structures of short stem and plug that are in the same location as the long stem/pilus in the piliated T4aPM, (3) the cluster_3 minor pilins/PilY1 are found in a ~1:1 ratio in T4aP, (4) minor pilins and PilY1 are essential for T4aP extension and T4aP-based motility, and (5) mutants in which the short stem and/or plug is disrupted, do not make pili. Based on these lines of evidence, we think that our conclusions are justified. We would be happy to discuss other possibilities but we cannot think of any one more likely.

- The pseudo-atomic model reliability is quite weak, being based on co-IP and speculations on interaction strengths. It is not clear to me what value this model adds to the paper, and I suggest to remove it.

Response: We absolutely agree with the reviewer that the homology models of the globular domains of the pilins may not be correct in all details. The value of these models is that they allows us to estimate the approximate volume these proteins could occupy in space. We believe that these models add significant value to the manuscript by confirming that the densities of short stem and plug together are sufficient to accommodate at least one copy of each of the

priming complex components (PilA, minor pilins, and PilY1.3) proposed here. This serves as a valid structural examination to support our conclusion drawn in the manuscript title. We note that while we have tried our best to arrange the pilin subunits in the model in a fashion that agrees with the results of our comprehensive biochemistry assay for protein-protein interactions as well as other available knowledge, we did not put emphasis on the arrangement nor any atomic details of the pilin subunits other than just filtered the overall model to 3 nm resolution and used its size to test and support the assignment of corresponding subtomogram average densities. However, we absolutely understand the reviewer's concern regarding that readers might overly interpret the information carried by the structural model. We have therefore toned down the language in the text to reflect the role of the model. Also, we now refer to the model of the priming complex as a "hypothetical structural model" (and not as a pseudo-atomic model). We moved the stepping-stones in the generation of the hypothetical structural model of the priming complex from a main text figure to Fig. S11a,b and now only include the model filtered to 3 nm resolution together with a component map of the T4aPM in Fig. 4b. Moreover, in Fig. S11a we clearly include the C-scores and we also write in the legend to Fig. S11a that "C-scores are expressed in numbers from -5 to +2, with +2 indicating high confidence". We have also toned down and corrected our language in the text to make it clear that the purpose of the structural model of the priming complex is to test whether the short stem and plug densities are feasible to accommodate all the proposed protein components (line 290-318).

- The analysis of the tips does not provide evidence that PilY1.3 resides there. While the mass-spec analysis suggests that this is the case, direct evidence is lacking. I wonder whether immunogold labelling could be done for cryo-tomography? This should allow identification of the ends of pili.

Response: We have added a new experiments (line 341-345 & Fig. S12b) in which we perform immuno-gold labeling on purified T4aP with a FLAG-tagged PilY1.3 protein. Indeed we observe T4aP-associated gold particles and not more than one particle per pilus. We have the same observations for cells in line 341-345 & Fig. S12c. However, the resolution of these images is too low to conclude that the gold particles are at the tips of pili. Therefore, we switch to cryo-ET. This allowed us to identify the specific, low density, kinked structure at the tip. The idea to do cryo-tomography on immune-gold labeled cells or purified pili is a very good one. But the frequency of this labeling is too low to make this a viable way forward. Based on the quantification and stoichiometry of cluster_3 minor pilins and PilY1 in purified pili, their importance for pilus formation, and the tip structure into which the filtered hypothetical structural model fits, we find that the evidence for the placement of the minor pilin/PilY1 complex at the pilus tip is compelling.

Reviewer #3 (Remarks to the Author):

In the present manuscript Treuner-Lange et al. show a very extensive study of the role of 3 different pilY1 homologues and minor pilins clusters in the dynamics of Type IV pilus in the gram-negative bacterium *Myxococcus xanthus*. Combinatorial genetic mutants of the 3 different clusters were combined with a wealth of techniques from biochemistry to structural cryo-EM to make a compelling case for the role of PilY1 and minor pilins in *Myxococcus xanthus* pili biogenesis. But starting with the title, not enough is done to explain the specificity of the *Myxococcus xanthus* model in terms of type IV pilus machinery and to contrast the present results with established results in other model systems. I will below summarize the points that I would wish to see addressed.

Response: Thank you very much for the encouraging comments. We changed the title as suggested below to make it clear that the manuscript focuses on the *M. xanthus* model system.

Major points:

- The dependence on pilQ to trigger the assembly of the type IV pilus machinery in an outside-in manner seems a specificity of Myxococcus. In Neisseria gonorrhoeae pilQ is not required for priming or elongation as a double pilQ+pilT mutant even shows polymerized pili accumulating in the periplasm (Wolfgang et al. EMBO Journal 19:6408-6418 (2000)).

Response: Thanks for pointing this out to us! And we apologize for this mistake and now state in line 126 and line 189 that "...assembly depends on the PilQ secretin in the OM in *M. xanthus*" and "localization was dependent on PilQ, which is essential for T4aPM assembly in *M. xanthus*".

- Still in Neisseria gonorrhoeae, using a similar FLAG tagging technique to the one presented in this manuscript it was shown that some minor pilins were confined to the periplasm (Imhaus et al. EMBO Journal 33:1767-1783 (2014)).

Response: Thanks for pointing this out to us and we apologize for this mistake. We have included this reference in the Introduction and now write in line 69 "Minor pilins in *P. aeruginosa* and *N. gonorrhoeae* have been detected in purified pili (ref, 5, 8, 10) and in the periplasm (ref 11).

- Because of the previous points and the changing roles played by pilY1 across type IV pili systems I think it is capital that the specificity of the Myxococcus xanthus model would be extensively discussed and in particular that the title would be amended along those lines: "PilY1 and minor pilins from a priming complex in the type Iva pilus machine and the pilus tip in the gram-negative bacterium Myxococcus xanthus"

Response: Thanks and done!

Minor points:

- The presence of pili on the outside of the cells body is a complicated result of the ability or not of priming growth but also the actual ratio of elongation and retraction dynamics. Because of this, the term pili formation can be vague both referring to its nucleation and further extension and existence in an extended state. In particular in line 108, the term T4aP formation could be replaced by T4aP presence in sheared fraction or something of the kind.

Response: Good point! We have changed the text throughout to say that a protein X is important for formation of T4aP if a strain lacking X does not make pili. Once we have demonstrated that the mutant even in the absence of the PilT retraction ATPase does not make pili, then we say that protein X is essential for pilus extension.

- Line 194: when discussing the involvement of pilQ in incorporating pilY1 maybe once again make sure that this is contrasted with the fact that pilQ is not necessary for this step in other systems.

Response: In line 276, we now included that this is the case in *M. xanthus*.

- The $\Delta 1\Delta 2$ cluster Δ pilT strain is well used as a control for pili presence in supplementary figure 3c, but that would be great if we could also have an image of the motility phenotype for the same strain added to this figure.

Response: We have included a new experiment with the T4aP-dependent motility phenotype of the $\Delta 1\Delta 2$ cluster Δ pilT strain in Fig. S6d.

- In supplementary figure 3b, there are some interesting changes in the amount of sheared pili obtained for different mutants. In particular, the relative ratio seems to be inverted between $\Delta 1\Delta 2$ and $\Delta 2\Delta 3$ when we go from the entire cluster being deleted to pilX being deleting (second and fourth lane from the right). It seems to indicate a complex interaction between the different proteins that could be interesting to point out.

Response: Supplementary figure 3b in the original submission is now Fig. 1d. We thank the reviewer for pointing this out to us. For this manuscript, we would like to focus on the role of minor pilins and PilY1 proteins in T4aP extension and their localization within the T4aPM and at the pilus tip. It is also clear from the label-free quantitative mass spectrometry (LFQ-MS) on whole cell extracts in Fig. S5 that there are some connection(s) between accumulation levels of cluster_3 and cluster_1 proteins. We hope to be able to work on these connections in the future.

REVIEWER COMMENTS

Reviewer #1 (Remarks to the Author):

The revised manuscript by Treuner-Lange et al. entitled "PilY1 and minor pilins form a priming complex in the type IVa pilus machine and the pilus tip in *Myxococcus xanthus*" is an improvement on the original submission and addresses many of my criticisms. As stated in my previous review, it is very likely that what the authors propose is true, that the minor pilins and PilY1 form a complex that primes pilus assembly and locates to the pilus tip. Their data support the priming/interaction claims (Figs 1-3) but such results are not particularly novel. The more novel aspect of the manuscript, that these proteins form a complex located at the pilus tip, a kinked tip no less (Figs 4-6), are not in my opinion sufficiently supported by the data.

The data shown in Fig. 4a are over-interpreted. What I see is that the density is virtually identical for the triple pilY1 mutant (column 2), the triple minor pilin mutant (column 4) and the mutant expressing cluster 3 only, with a variant PilY1.3, pilY1.3 Δ vWFA-FLAG (column 7). In all of these sub-tomogram averages the stem and plug densities are absent. Since deleting the minor pilins and deleting PilY1 both result in loss of density its not clear which density corresponds to which proteins. And since the Δ 1 Δ 2cluster Δ pilY1.3 mutant complemented with pilY1.3 Δ vWFA-FLAG lacks density for both the stem and the plug, suggesting that the pilins and PilY1.3 Δ vWFA-FLAG are absent in this machine, this sub-tomogram image does not support the argument that the C-terminal region of PilY1 is part of the stem and the vWFA domain is the plug. The PilY1.3 Δ vWFA-FLAG variant doesn't appear to associate with the pilus machinery despite the data suggesting it interacts with minor pilins. Does it localize to the poles of the bacterium?

Why isn't the Δ 1 Δ 2cluster Δ pilY1.3 mutant shown on its own and expressing WT PilY1.3 in Fig. 4a? These seem like important controls.

Lines 230-232 –The authors attribute the lower part of the stem density to the major pilin and the upper part to the minor pilins and PilY1. But the upper part of the stem density is present in the pilA mutant. Are they suggesting that the stem/plug density observed in this mutant are minor pilins and PilY1? If so, how can the minor pilins be suspended in space and not connected to the inner membrane? This does not seem plausible. Or is all of the stem/plug density shown in the pilA mutant image (column 3) attributed to PilY1? That doesn't seem to be the case according to the model presented in Fig. 4b.

Line 310-311 and Fig. 4b and Supplementary Fig. 11. It is troubling that the densities for the PilY1 domains are so small. The PilY1 N-terminal domain is ~95 kDa (~860 amino acids) and the C-terminal domain is ~60 kDa (~540 aa) yet both are modeled as if they are smaller than PilW3, which is ~44 kDa. The authors argue that more than half of the protein is flexible and therefore not visible in the sub-tomogram averages. While this is possible it seems unlikely.

Overall, the subtomogram averages and computational models do not provide convincing data for the model presented in Fig. 4b.

The modeling of the minor pilins and PilY1 at the kinked tip is also unconvincing. No data are presented to support the localization of PilY1 or the minor pilins at the pilus tip, or that the kinked tips are in fact the true tip of the pilus and not just broken pili. The tips look different in every image shown in Sup. Fig. 13. The volumes they show for minor pilins/PilY1 could be made to fit into any of these as well as a straight tip.

Minor points:

Line 240: "remove "deleted for"

Line 280-283: This statement is confusing. The authors conclude here that the short stem and plug, which they attribute to the minor pilins and PilY1, are integral parts of the non-piliated T4aPM and not incorporated as part of the PilB-stimulated pilus extension process. I understand what they mean – that these components pre-assemble instead of being added as the pilus grows - but it seems to suggest that they are part of the machine and not the pilus. This should be reworded.

Line 377-378: The authors propose that the same priming complex reestablished every time after pilus retraction. Do they actually mean “remains intact”?

Line 388: The authors throw out an alternative explanation for the “stop-retraction mechanism” that is not well explained. It is also unlikely that the N-terminal α -helices of the minor pilins are packed differently than those of the major pilin given the sequence conservation.

Reviewer #2 (Remarks to the Author):

I am satisfied that the authors responded exhaustively to my comments on the first submission, and support publication of the revised version.

I'd like to comment on their statement in the answers to referees that: "A subtomogram average is a result of retaining consistent densities and removing inconsistent densities among all molecules used in the average, so all densities revealed are considered significant. ".

A flawed comment and a bit presumptuous: if this were true, there would be no need for validation, nor of resolution measurements.

Of course, there are many reasons why a subtomogram average can be wrong - globally or locally - and several examples of this too.

Specifically for this case, in a difference map there will be a number of densities, and it is important to establish that the density attributed to GFP is significant compared to the noise. The authors have done that, so I'm happy, despite their comment above.

Reviewer #3 (Remarks to the Author):

The responses to the comments from the reviewers were thorough and the new text and extra experiments went a great way towards clarifying the message of the manuscript. While most likely not amounting to a preclusion of publication, there are still a few points that I would want to see addressed.

1. The role of pilY1 (pilC in Neisseria terminology) in Neisseria species is far from settled with roles both in adhesion and pili dynamics. There are different studies reporting piliation not correlating with the amount of pilY1 or even in its absence. I think that would be worth mentioning in the text, maybe around line 80. I have added here two such references.

Functional Implications of the Expression of PilC Proteins in Meningococci
Virji et al. Mol Microbiol 1995 Jun;16(6):1087-97

Pilus biogenesis and epithelial cell adherence of Neisseria gonorrhoeae pilC double knock-out mutants. Rudel et al. Molecular Microbiology, 31 Aug 1995, 17(6):1057-1071

2. The crucial mechanistic role that the authors assigned to the vWFA domain of PilY1.3 in their hypothetical model seems to limit the applicability of the model to many other systems including the Myxococcus cluster 1 and 2 as, as pointed out by the authors, those proteins do not possess such domains as in PilY1 proteins in other bacterial systems. So the mechanistic understanding of these other systems ought to be different. I think it is important that this point be touched upon in the discussion.

3. The absence of the kink in 30% of the pili could simply be due to the fact that there are more than one mechanism for priming or nucleation of the pili. Relative ratios of the different mechanisms might depend on the specific Type IV pilus system and maybe the exact circumstances. While not removing anything from the discoveries made by the authors, acknowledging the possibility of different priming mechanisms will enable the entire data of this manuscript and the literature to come nicely together.

Typos: Line 438 most likely centrifuged instead of harvested.

REVIEWER COMMENTS

Reviewer #1 (Remarks to the Author):

The revised manuscript by Treuner-Lange et al. entitled “PilY1 and minor pilins form a priming complex in the type IVa pilus machine and the pilus tip in *Myxococcus xanthus*” is an improvement on the original submission and addresses many of my criticisms. As stated in my previous review, it is very likely that what the authors propose is true, that the minor pilins and PilY1 form a complex that primes pilus assembly and locates to the pilus tip. Their data support the priming/interaction claims (Figs 1-3) but such results are not particularly novel. The more novel aspect of the manuscript, that these proteins form a complex located at the pilus tip, a kinked tip no less (Figs 4-6), are not in my opinion sufficiently supported by the data.

The data shown in Fig. 4a are over-interpreted. What I see is that the density is virtually identical for the triple pilY1 mutant (column 2), the triple minor pilin mutant (column 4) and the mutant expressing cluster 3 only, with a variant PilY1.3, pilY1.3ΔvWFA-FLAG (column 7). In all of these sub-tomogram averages the stem and plug densities are absent. Since deleting the minor pilins and deleting PilY1 both result in loss of density its not clear which density corresponds to which proteins. And since the Δ1Δ2clusterΔpilY1.3 mutant complemented with pilY1.3ΔvWFA-FLAG lacks density for both the stem and the plug, suggesting that the pilins and PilY1.3ΔvWFA-FLAG are absent in this machine, this sub-tomogram image does not support the argument that the C-terminal region of PilY1 is part of the stem and the vWFA domain is the plug.

Response: We understand the reviewer’s argument and agree that the arrangement of minor pilins and PilY1 in the priming complex cannot be done based on the subtomogram averages only. Similarly, we agree that assignment of the C-terminal region of PilY1 to the short stem and the vWFA domain to the plug cannot be done solely based on the subtomogram averages. Therefore, we write in line 237-238 “Because minor pilins and PilY1.3 are mutually dependent for incorporation into the T4aPM, their arrangement within the T4aPM was less clear”. In the following paragraphs, we then dissect the arrangements of these proteins.

In the revised manuscript, we have now carefully rewritten and described in more details the logic that we followed in these experiments and how we combined different experimental approaches to map out the arrangement of PilA, minor pilins and PilY1. Briefly,

- 1) We show that PilY1.3 interacts with minor pilins (lines 170-182 & 202-213; Fig. 3), and minor pilins interact with PilA (lines 148-168; Fig. 2).
- 2) We show that minor pilins and PilY1.3 are mutually dependent for stability but independent of PilA (lines 193-201; Fig. 3).
- 3) We show that PilY1.3 and minor pilins incorporation into the T4aPM is mutually dependent supporting that they incorporate as a complex, and their incorporation is independent of PilA (lines 193-201).
- 4) We show that subtomogram averages of T4aPM in *M. xanthus* strains deleted for all *pilY1* (Δ1Δ2Δ3 *pilY1*) or nine minor pilins (Δ9 minor pilins) lack the entire short stem and plug but in a strain deleted *pilA* lacks only the lower part of the short stem (lines 220-231; Fig. 4). Because PilY1.3 and minor pilins incorporation into the T4aPM are independent of PilA (see above), these results suggest that the lower part of the stem is accounted for by PilA and the upper part of the short stem and the plug comprise the complex of minor pilins and PilY1.
- 5) Because the plug density (~4 nm in diameter) is not big enough to fit the entire PilY1.3 (Mw ~160 kDa; theoretical diameter is ~7 nm if calculated as a single globular protein), we reasoned that part of PilY1.3 is likely in the short stem. Importantly, PilY1.3 contains two globular domains separated by regions of undefined structure (lines 240-250).
- 6) Fusion of a sfGFP to the PilY1.3 C-terminus gives rise to an additional density on the tip of the short stem in the T4aPM subtomogram average (Fig. 4), which also suggests the location of PilY1.3 C-terminal portion to be in the short stem (lines 252-270).

7) In new experiments, we show that a PilY1.3 Δ vWFA variant is incorporated in to the T4aPM (although probably less effectively than the full-length protein); and, in pull-down experiments with full-length PilY1.3 and PilY1.3 Δ vWFA they show the same interactions with minor pilins as well as with PilO in the lower-periplasmic ring (lines 270-287; Figs. 3, S9), further supporting the remaining C-terminal portion of PilY1.3 Δ vWFA is the part of PilY1 that locates in the short stem. 8) In the subtomogram average of the PilY1.3 Δ vWFA mutant, T4aPM lacked the short stem and plug densities while the lower periplasmic ring was intact (lines 289-297; Fig. 4).

Together, the results from 1-8 prompted us to suggest a model whereby the PilY1.3 N-terminal vWFA domain is in the plug and the C-terminal PilY1 domain in the short stem (lines 297-300). Moreover, combining all these data, the simplest model that agrees with all the available data is one in which the short stem and plug densities represent a priming complex for T4aP formation composed of, from the inner membrane outwards, PilA, the minor pilins and PilY1 and in which the PilY1.3 N-terminal vWFA domain accounts for the plug and C-terminal PilY1 domain is at the tip of the short stem (lines 308-329).

Subsequently, we challenged this model and tested for its feasibility by constructing a hypothetical structural model (lines 330-363). We now make very clear in line 332-334 that “the goal of this model was not generating a precise structural model of the priming complex but to test the feasibility of the suggested arrangement of components”. We show that the size of a complex of PilA/minor pilins/PilY1.3-C-terminal domain fits to the short stem density and the size of the PilY1.3-vWFA domain fits to the plug density at 3 nm resolution (Figs. 4b, S11). Therefore, we write “We conclude that the hypothetical structural model supports the feasibility of the suggested arrangement of components in the T4aPM” (lines 359-361).

Protein sequence analysis among PilY1 family proteins also reveals their C-terminal portion including the PilY1 domain is more conserved comparing to the N-terminal portion (Fig. S2), again supporting it is the C-terminal portion interacting with the also conserved minor pilins and located in the short stem.

The PilY1.3 Δ vWFA-FLAG variant doesn't appear to associate with the pilus machinery despite the data suggesting it interacts with minor pilins.

Does it localize to the poles of the bacterium?

Response: Point well-taken! In the revised manuscript, we included new experiments in which we generated and analyzed a PilY1.3 Δ vWFA-sfGFP variant. We found that this variant localizes polarly in a manner that depends on PilQ, thus, strongly supporting that it is incorporated into the T4aPM (lines 276-280). Moreover, based on the pull-down experiments, PilY1.3 Δ vWFA-FLAG interacts with the minor pilins as well as with PilO in the lower-periplasmic ring (lines 280-285).

Finally, in the revised manuscript, we describe the subtomogram average of the PilY1.3 Δ vWFA-FLAG mutant more carefully. Specifically, we find that the T4aPM structure lacked the plug and short stem densities while the lower periplasmic ring was intact (Fig. 4a, 7th column; lines 287-297). Because the intactness of the lower-periplasmic ring depends on PilY1, the subtomogram average also supports that PilY1.3 Δ vWFA-FLAG is incorporated in to the T4aPM.

Why isn't the Δ 1 Δ 2cluster Δ pilY1.3 mutant shown on its own and expressing WT PilY1.3 in Fig. 4a? These seem like important controls.

Response: The suggested Δ 1 Δ 2cluster Δ pilY1.3 mutant on its own corresponds to the imaged Δ 1 Δ 2 Δ 3 pilY1 strain, which is included in Fig 4a (column 2). The second suggested strain corresponds to the Δ 1 Δ 2cluster strain, which is included in Fig. 4a (column 5). We did not do additional cryo-ET experiments on the Δ 1 Δ 2cluster Δ pilY1.3 mutant ectopically complemented

with native PilY1.3 because that strain behaves like WT for T4aP-dependent motility (Supplementary Figure 6a).

Lines 230-232 –The authors attribute the lower part of the stem density to the major pilin and the upper part to the minor pilins and PilY1. But the upper part of the stem density is present in the pilA mutant. Are they suggesting that the stem/plug density observed in this mutant are minor pilins and PilY1?

Response: Yes (and please see below for our reasoning).

If so, how can the minor pilins be suspended in space and not connected to the inner membrane?

This does not seem plausible.

Response: Thanks for pointing this out to us. Based on our data from the BACTH (lines 164-166), the N-terminal segment of the PilA N-terminal α -helix is important for the interaction of PilA with minor pilins. Because the N-terminal α -helices of pilins are tightly associated helically to form the core of the T4P fiber and, when considering the packing of pilins in T4aP is a right-handed 1-start helix, each pilin subunit ($\#n$) interacts with the fourth ($\#n+3$) and the fifth ($\#n+4$) pilin subunits down the fiber. In our model of the priming complex, there are five pilin subunits: PilA at the bottom, followed by 4 minor pilins PilV3, FimU3, PilW3, PilX3. With the absence of PilA at the bottom, it is possible that the remaining 4 minor pilins cannot form a complete helical turn using their N-terminal α -helices, but instead ‘float’ in the center of the lower periplasmic ring, stabilized only by interactions with PilY1 and PilO. The density of their randomly positioned N-terminal α -helices would be averaged out in the subtomogram average and therefore appear as a gap between the IM and the remaining stem. In the revised manuscript, we have now included a brief discussion about this point (lines 321-327) “We also speculate that in the T4aPM subtomogram average of the $\Delta pilA$ mutant (Fig. 4a, 3rd column), the clear gap between the remaining short stem and the IM support that in the absence of PilA the minor pilins likely do not properly interact with one another but instead “float” in the region with their N-terminal α -helix in the IM and globular head domains near the lower-periplasmic ring to interact with other T4aPM components including PilY1. Their random locations therefore caused a decrease of the averaged density at the lower part of the short stem”.

Or is all of the stem/plug density shown in the pilA mutant image (column 3) attributed to PilY1? That doesn't seem to be the case according to the model presented in Fig. 4b.

Response: Our data show that PilY1 interacts with the minor pilins and their incorporations into the T4aPM are mutually dependent but independent of PilA (lines 193-201). Because we previously showed that the $\Delta pilA$ mutant incorporates the remaining core proteins and we show here that the $\Delta pilA$ mutant incorporates minor pilins (using PilW3-sfGFP as a proxy) as well as PilY1.3-sfGFP into the T4aPM, we prefer our current assignment that the remaining stem/plug density comprise the minor pilins and PilY1.3. In the revised text, we have rewritten to make this clearer (lines 235-236)

Line 310-311 and Fig. 4b and Supplementary Fig. 11. It is troubling that the densities for the PilY1 domains are so small. The PilY1 N-terminal domain is ~95 kDa (~860 amino acids) and the C-terminal domain is ~60 kDa (~540 aa) yet both are modeled as if they are smaller than PilW3, which is ~44 kDa. The authors argue that more than half of the protein is flexible and therefore not visible in the sub-tomogram averages. While this is possible it seems unlikely.

Response: We apologize for this confusion. The size of the PilY1 domain in the original figures seemed small in the model simply because the minor pilins were blocking the view in the images. We apologize for this, and we have rotated the model in the Figs. 4b and S11 for a better view accordingly.

Also, we apologize for not being clearer about which parts of the proteins were modelled in Supplementary Figure 11a. We have now included in this figure precise mentioning of the coordinates of the domains modelled. Specifically, the modelled N-terminal vWFA domain of PilY1.3 is 263 aa in length (28.1 kDa) and the C-terminal PilY domain is 421 aa in length (45.8 kDa). The latter is only slightly bigger than the globular domain of PilW3, which is 35.7 kDa (329 aa) in size. We added the missing information about the sizes in the revised Supplementary Fig. 11a.

Overall, the subtomogram averages and computational models do not provide convincing data for the model presented in Fig. 4b.

Response: As discussed above, the component map presented in Fig. 4b was not derived solely based on the subtomogram averages. Please see the response above.

Also, we now describe more clearly that the goal of constructing the hypothetical structural model was to test for the feasibility of the order of components in the priming complex. We have now made this clearer in line 330-334 by writing that “As a further test of our model for the arrangement of PilA, minor pilins and PilY1 in the priming complex, we took advantage of available structural information for homologous proteins to generate a hypothetical structural model of the priming complex using the cluster_3 proteins. Of note, the goal of this model was not generating a precise structural model of the priming complex but to test the feasibility of the suggested arrangement of components”. Later we state that “We conclude that the hypothetical structural model supports the feasibility of the suggested arrangement of components in the T4aPM” (lines 359-361). Therefore, we believe that the hypothetical structural model is very helpful as a test of the overall arrangement of components in the priming complex.

We also clearly acknowledge that it will an important future goal to elucidate the exact structure of the priming complex “...the exact structure of the globular domains of the pilins, their precise order and packing, as well as the stoichiometry of the involved proteins remain to be revealed by future studies” in line 361-363.

The modeling of the minor pilins and PilY1 at the kinked tip is also unconvincing. No data are presented to support the localization of PilY1 or the minor pilins at the pilus tip, or that the kinked tips are in fact the true tip of the pilus and not just broken pili. The tips look different in every image shown in Sup. Fig. 13. The volumes they show for minor pilins/PilY1 could be made to fit into any of these as well as a straight tip.

Response: We have presented mass spectrometry data supporting the cluster 3 minor pilins and PilY1.3 form a complex with ~1 copy per purified pilus (lines 370-384). Because the function of a priming complex would most likely be to serve as a “scaffold” for initiating PilA incorporation into the growing pilus, we hypothesized that the priming complex would exit from the T4aPM and localize at the tip of the extended pilus. To test this hypothesis, we collected and examined cryo-electron tomograms of intact cells and found more than 70% of the observed pilus tips showed a kinked structure. Inspired by the reviewer’s comment, we have now carefully measured the length and width of all the observed kinked structures as well as kinked angles as listed in Fig. S11 and included in Fig. 5c. We then speculated (lines 396-404) that the kinked structure is the parts of PilY1 with discontinued density to the short stem (i.e., in the plug) and/or invisible due to flexibility in the subtomogram average (e.g., in between the plug and short stem) in the T4aPM. We also speculated that these parts of PilY1.3 would be able to dangle around at the pilus tip after the stem/pilus extends out from the cell resulting in the kinked structures, while in the T4aPM these parts of PilY1.3 would be more aligned because of interactions with the remaining T4aPM components. We find that based on its dimensions, the kinked structure is capable of accommodating the PilY1.3 vWFA domain model (~4 nm in diameter) and more. Moreover, in some of the kinked structures a distinct globular density can

be seen, and it connects to the tip of the pilus shaft through a lighter density (Fig. 5c-iv). The size of the globular density and its distance from the tip of the pilus shaft can match well to the size of the plug in the T4aPM and its distance from the tip of the short stem (Fig. S11c,d). To make these comparisons clearer, we have made a new figure for the side-by-side comparison of their densities size measurements as well as fitting with the hypothetical structural model (Fig. S11). Altogether, these results support our hypothesis of the priming complex being on the tip of pilus. We now describe more carefully the logic that we use to compare the kinked tip structure and priming complex in the T4aPM (lines 396-413).

To be more conservative on the argument of the priming complex being at the tip of the extended pilus, we have changed the title of the revised manuscript to not include “pilus tip” and also moved the fitting of the hypothetical structural model of the priming complex at the pilus tip from the main Fig. 5c to Fig. S11c,d.

The densities of the kinked structure, as the reviewer pointed out, vary in dimensions. If the kinked density is indeed from the priming complex, its polymorphism could be due to 1) flexibility within the PilY1 protein, 2) pilus tips were frozen and viewed in different orientations in the tomograms, and/or 3) differences among the three minor pilin/PilY1 clusters in *M. xanthus*. Concerning the latter possibility, we consider this unlikely because WT cells mostly synthesize minor pilins and PilY1 from cluster 3 (Fig. S5). However, we also agree that it is possible that it is caused by the breakage of a pilus. We have therefore discussed in the main text why we believe the kinked density is more likely caused by the presence of the priming complex but not the breakage of pilus (lines 415- 420) “At this point, we could not rule out the possibility that some of the pilus tips observed were from snapped pili due to their surface binding and retraction. However, since the cells used for these analyses were cultured in suspension (Methods) with limited chance for surface binding before being frozen directly for cryo-ET imaging, we speculate that if there are broken pili, they would only account for a minority of pili tips in the whole population and not represent the kinked structure in 70% of observed pili.” We have also softened our final conclusion about the kinked tip to “Thus, these observations are in agreement with the notion that the kinked structure at the pilus tip may represent the priming complex with the PilY1.3 C-terminal domain at the pilus proximal end of the kink and the N-terminal vWFA domain at the ultimate tip (Supplementary Fig. 11c,d)” (lines 410-413).

Minor points:

Line 240: “remove “deleted for”

Response: Thanks & done!

Line 280-283: This statement is confusing. The authors conclude here that the short stem and plug, which they attribute to the minor pilins and PilY1, are integral parts of the non-piliated T4aPM and not incorporated as part of the PilB-stimulated pilus extension process. I understand what they mean – that these components pre-assemble instead of being added as the pilus grows - but it seems to suggest that they are part of the machine and not the pilus. This should be reworded.

Response: We revised the text to as suggested (lines 314-3317).

Line 377-378: The authors propose that the same priming complex reestablished every time after pilus retraction. Do they actually mean “remains intact”?

Response: Thank you for the suggestion. We have revised the text as suggested (line 432).

Line 388: The authors throw out an alternative explanation for the “stop-retraction mechanism” that is not well explained. It is also unlikely that the N-terminal α -helices of the minor pilins are

packed differently than those of the major pilin given the sequence conservation.

Response: Thanks for the comment. We have revised the text to add more details: (lines 438-440) "Non-canonical arrangement of the N-terminal α -helices of the minor pilins in the priming complex caused by the packing of their varied globular domains and/or the binding with PilY1 is another possible mechanism."

Reviewer #2 (Remarks to the Author):

I am satisfied that the authors responded exhaustively to my comments on the first submission, and support publication of the revised version.

I'd like to comment on their statement in the answers to referees that: "A subtomogram average is a result of retaining consistent densities and removing inconsistent densities among all molecules used in the average, so all densities revealed are considered significant. "

A flawed comment and a bit presumptuous: if this were true, there would be no need for validation, nor of resolution measurements.

Of course, there are many reasons why a subtomogram average can be wrong - globally or locally - and several examples of this too.

Specifically for this case, in a difference map there will be a number of densities, and it is important to establish that the density attributed to GFP is significant compared to the noise. The authors have done that, so I'm happy, despite their comment above.

Response: Thank you very much! The useful advice is well taken.

Reviewer #3 (Remarks to the Author):

The responses to the comments from the reviewers were thorough and the new text and extra experiments went a great way towards clarifying the message of the manuscript. While most likely not amounting to a preclusion of publication, there are still a few points that I would want to see addressed.

Response: Thank you very much!

1. The role of pilY1 (pilC in *Neisseria* terminology) in *Neisseria* species is far from settled with roles both in adhesion and pili dynamics. There are different studies reporting piliation not correlating with the amount of pilY1 or even in its absence. I think that would be worth mentioning in the text, maybe around line 80. I have added here two such references.

Functional Implications of the Expression of PilC Proteins in Meningococci

Virji et al. Mol Microbiol 1995 Jun; 16(6):1087-97

Pilus biogenesis and epithelial cell adherence of *Neisseria gonorrhoeae* pilC double knock-out mutants. Rudel et al. Molecular Microbiology, 31 Aug 1995, 17(6):1057-1071

Response: Thanks for pointing us in the direction of these two papers. We note that in the Virji et al (1995) paper, the authors report on clinical isolates of *N. meningitidis* and find that the levels of PilC (=PilY1) and pilin do not correlate. From reading the paper, it was not clear to us whether the authors looked at both PilC1 and PilC2. Moreover, it was not clear to us whether the authors reported on total levels of proteins in cell extracts or whether the reported levels are in purified pili. Therefore, we would prefer not to include a reference to this paper. We note that in the Rudel et al. (1995) paper, the authors report that a double pilC1 pilC2 mutant of *N. gonorrhoeae* is non-piliated. Subsequently, they isolate spontaneous suppressor mutants that are piliated. The suppressors contain a 70 kDa protein of unknown function, which the authors suggest replaces the function of PilC. The identity of the 70 kDa protein is not known. Given that it is unclear what is happening in the suppressors, we would prefer not to include a reference to this paper. We would like to add that it is a possibility that PilY1 may have species-specific functions in addition to those that we report for *M. xanthus*. We do not exclude that possibility in

our manuscript. However, for this manuscript we would like to focus on the importance of PilY1 and minor pilins in T4aP formation. We would also like to add that in line 82, we state that "...PilY1 is important for T4aP formation..." to cover the PilY1 literature. Subsequently, we zoom in on *M. xanthus* and the function of PilY1 in T4aP formation.

2. The crucial mechanistic role that the authors assigned to the vWFA domain of PilY1.3 in their hypothetical model seems to limit the applicability of the model to many other systems including the Myxococcus cluster 1 and 2 as, as pointed out by the authors, those proteins do not possess such domains as in PilY1 proteins in other bacterial systems. So the mechanistic understanding of these other systems ought to be different. I think it is important that this point be touched upon in the discussion.

Response: One of the fascinating aspects of PilY1 proteins is their diverse N-terminal domains. In line 451-455 we write that "Placing the variable N-terminal domain of PilY1 at the extreme pilus tip rationalizes how it might provide host cell specificity during infections, as suggested (ref. 16, 18, 20, 36, 37). We speculate that the three separate PilY1/minor pilin gene clusters in *M. xanthus* enable the generation of T4aP tips with different properties, facilitating recognition of different substrates". We believe – and hope that you agree with us – that these sentences nicely cover the notion that the different PilY1 proteins at the pilus tip may provide specificity for different substrates.

3. The absence of the kink in 30% of the pili could simply be due to the fact that there are more than one mechanism for priming or nucleation of the pili. Relative ratios of the different mechanisms might depend on the specific Type IV pilus system and maybe the exact circumstances. While not removing anything from the discoveries made by the authors, acknowledging the possibility of different priming mechanisms will enable the entire data of this manuscript and the literature to come nicely together.

Response: Thank you for pointing out this possibility to us. We have carefully considered this comment. However, it is clear that in *M. xanthus* T4aP formation crucially depends on minor pilins and PilY1. So, we would prefer not to discuss a different priming mechanism, which would include unknown proteins.

Typos: Line 438 most likely centrifuged instead of harvested.

Response: Thanks & changed as suggested!

REVIEWERS' COMMENTS:

Reviewer #1 (Remarks to the Author):

The author's efforts to address the concerns of the reviewers are laudable. While I remain unconvinced that the conclusions are fully supported by the data, the manuscript nonetheless provides an important framework for understanding pilus priming and should be seen and judged by a broad readership. I therefore support its publication.

Lisa Craig

REVIEWERS' COMMENTS:

Reviewer #1 (Remarks to the Author):

The author's efforts to address the concerns of the reviewers are laudable. While I remain unconvinced that the conclusions are fully supported by the data, the manuscript nonetheless provides an important framework for understanding pilus priming and should be seen and judged by a broad readership. I therefore support its publication.

Lisa Craig

Response: Dear Lisa Craig,
Thanks for all your inspiring comments.